# Neurogenetic dissection of the *Drosophila* lateral horn reveals major outputs, diverse behavioural functions, and interactions with the mushroom body

Michael-John Dolan[1,2], Shahar Frechter[2], Alexander Shakeel Bates[2], Chuntao Dan[1], Paavo Huoviala[2], Ruairí JV Roberts[3], Philipp Schlegel[2,3], Serene Dhawan[2,3], Remy Tabano[1], Heather Dionne[1], Christina Christoforou[1], Kari Close[1], Ben Sutcliffe[2], Bianca Giuliani[1], Feng Li[1], Marta Costa[3], Gudrun Ihrke[1], Geoffrey Wilson Meissner[1], Davi D Bock[1], Yoshinori Aso[1], Gerald M Rubin[1†]*, Gregory SXE Jefferis[1,2,3†]*

[1]Howard Hughes Medical Institute, Janelia Research Campus, Ashburn, United States; [2]Division of Neurobiology, MRC Laboratory of Molecular Biology, Cambridge, United Kingdom; [3]Department of Zoology, University of Cambridge, Cambridge, United Kingdom

*For correspondence:
rubing@janelia.hhmi.org (GMR);
jefferis@mrc-lmb.cam.ac.uk
(GSXEJ)

†These authors contributed
equally to this work

Competing interests: The
authors declare that no
competing interests exist.

Reviewing editor: K
VijayRaghavan, National Centre
for Biological Sciences, Tata
Institute of Fundamental
Research, India

**Abstract** Animals exhibit innate behaviours to a variety of sensory stimuli including olfactory cues. In *Drosophila*, one higher olfactory centre, the lateral horn (LH), is implicated in innate behaviour. However, our structural and functional understanding of the LH is scant, in large part due to a lack of sparse neurogenetic tools for this region. We generate a collection of split-GAL4 driver lines providing genetic access to 82 LH cell types. We use these to create an anatomical and neurotransmitter map of the LH and link this to EM connectomics data. We find ~30% of LH projections converge with outputs from the mushroom body, site of olfactory learning and memory. Using optogenetic activation, we identify LH cell types that drive changes in valence behavior or specific locomotor programs. In summary, we have generated a resource for manipulating and mapping LH neurons, providing new insights into the circuit basis of innate and learned olfactory behavior.
DOI: https://doi.org/10.7554/eLife.43079.001

## Introduction

Understanding how neural circuits transform sensory input into behaviour is a fundamental challenge in neuroscience. The ability to manipulate anatomically and functionally related groups of neurons (known as cell types) through which this sensory information flows is an essential prerequisite to understanding this problem (*Luo et al., 2018*). Olfaction is an ideal sensory system to address this question. The olfactory system is only one synapse from the sensory periphery to the central brain, with common anatomical and circuit logic across the animal kingdom (*Su et al., 2009*). This makes the the tractable and manageably small *Drosophila melanogaster* brain an attractive model for the processing of olfactory information. After the first processing relay (the insect antennal lobe or mammalian olfactory bulb) the olfactory system diverges into at least two pathways (*Marin et al., 2002*; *Miyamichi et al., 2011*; *Sosulski et al., 2011a*; *Wong et al., 2002*): A hardwired, chemotopic circuit that is thought to mediate innate behaviour (the insect lateral horn and mammalian cortical amygdala) and a decorrelated, unstructured circuit that is required for memory storage and retrieval (insect mushroom body or mammalian piriform cortex) (*Choi et al., 2011*; *Dubnau et al., 2001*;

*Gruntman and Turner, 2013*; *Heimbeck et al., 2001*; *McGuire et al., 2001*; *Parnas et al., 2013*; *Root et al., 2014a*; *Sacco and Sacchetti, 2010*). The development of cell type specific neurogenetic tools has led to significant insights into how the *Drosophila* mushroom body subserves learning and memory, but the form and function of innate circuits remain little known in both insects and mammals (*Aso et al., 2014a*; *Aso et al., 2014b*; *Claridge-Chang et al., 2009*; *Dubnau et al., 2001*; *Krashes et al., 2007*; *Liu et al., 2012*; *Root et al., 2014a*; *Tanaka et al., 2008*). In mice the cortical amygdala is both necessary and sufficient for innate olfactory responses; however, the neurons that drive avoidance and attraction intermingle (*Root et al., 2014a*) and the detailed cell types and their anatomy and function are unknown.

*Drosophila* olfactory processing begins at the antennae where receptors on sensory neurons bind odorant molecules and transduce this interaction into action potentials (*Masse et al., 2009*). This information is transmitted into the central brain to the antennal lobe (AL), a region consisting of ~50 subregions termed glomeruli (*Couto et al., 2005*; *Fishilevich and Vosshall, 2005*). Sensory neurons expressing the same receptor converge to the same glomerulus, an organizational feature shared with the mammalian brain (*Masse et al., 2009*; *Su et al., 2009*). In the AL, olfactory information is reformatted and relayed deeper into the brain by projection neurons (PNs), the output cells of the AL. Uniglomerular PNs (mPN1s) have their dendrites in a single glomerulus and send axons to the two higher olfactory brain regions, the mushroom body (MB) and the lateral horn (LH) (*Figure 1A*) (*Jefferis et al., 2007*; *Marin et al., 2002*; *Wong et al., 2002*). Several other classes of PNs exist, such as the oligoglomerular inhibitory PNs with dendrites in several glomeruli which project solely to the LH (*Tanaka et al., 2012*) and provide olfactory-induced inhibition to LH neurons (*Liang et al., 2013*; *Parnas et al., 2013*; *Strutz et al., 2014*; *Wang et al., 2014*).

Although they receive the same input from mPN1s, the MB and LH have distinct connectivity rules. The axons of the PNs connect randomly with different subsets of Kenyon cells, the intrinsic neurons of the MB (*Caron et al., 2013*; *Eichler et al., 2017*; *Gruntman and Turner, 2013*). This results in sparse olfactory representations which are additionally maintained by local MB inhibition (*Papadopoulou et al., 2011*). In contrast, PN-to-LH connectivity is stereotyped and the same PNs connect to the same LH neurons across different animals (*Fişek and Wilson, 2014*; *Frechter et al., 2019*; *Jeanne et al., 2018*; *Jefferis et al., 2007*; *Kohl et al., 2013*). An analysis of single-cell data revealed the axonal domains of PNs in the LH intermingle, with a demarcation between pheromonal and non-pheromonal PNs in the ventral and dorsal LH respectively (*Jefferis et al., 2007*). LH neurons receive synaptic input from PNs of different glomeruli and some sum their input linearly (*Fişek and Wilson, 2014*; *Frechter et al., 2019*; *Jeanne et al., 2018*). In response to odours, LH neurons are more broadly tuned than their inputs but able to categorise odours by chemical class (*Frechter et al., 2019*).

Given the distinct anatomical and physiological properties of these higher brain regions, it is unsurprising that their neurons have different functions in olfactory behaviour. Direct functional manipulations have almost exclusively focused on the MB pathway. By spatiotemporally silencing Kenyon cells, many studies have demonstrated that MB neurotransmission is necessary for the formation, consolidation and retrieval of olfactory memories but dispensable for innate olfactory responses (*Dubnau et al., 2001*; *Heimbeck et al., 2001*; *Krashes et al., 2007*; *McGuire et al., 2001*; *Parnas et al., 2013*), indirectly implicating the LH in innate olfactory processing. Recent cell type specific neurogenetic tools have allowed for the fine dissection of both MB structure and function (*Aso et al., 2014a*; *Tanaka et al., 2008*). Kenyon cell axons are compartmentalized into 15 distinct units that each receive dopamine modulation from dopaminergic neurons (DANs) and the dendrites of Mushroom Body output neurons (MBONs) which project to different brain regions to mediate valence behaviour (ie. approach or aversion) (*Aso et al., 2014a*; *Aso et al., 2014b*; *Owald et al., 2015*). During learning, dopaminergic input to distinct compartments, coincident with olfactory stimulation results in synaptic depression of the KC-to-MBON synapse, producing a learned response by altering the net activity of the MBON population, which control attraction or approach (*Aso et al., 2014b*; *Cohn et al., 2015*; *Hige et al., 2015a*).

In contrast to the MB, there have been no studies to date manipulating single or small constellations of identified cell types of the adult LH during olfactory behaviour, due to a lack of reagents available to control LH cell types. Indirect evidence, such as lesioning the MB, implicates the LH neuropil in innate olfactory responses such as stereotyped motor behaviour in response to olfactory stimulation (*Jung et al., 2015*; *Lin et al., 2013*; *Min et al., 2013*; *Suh et al., 2004*) or pheromone-

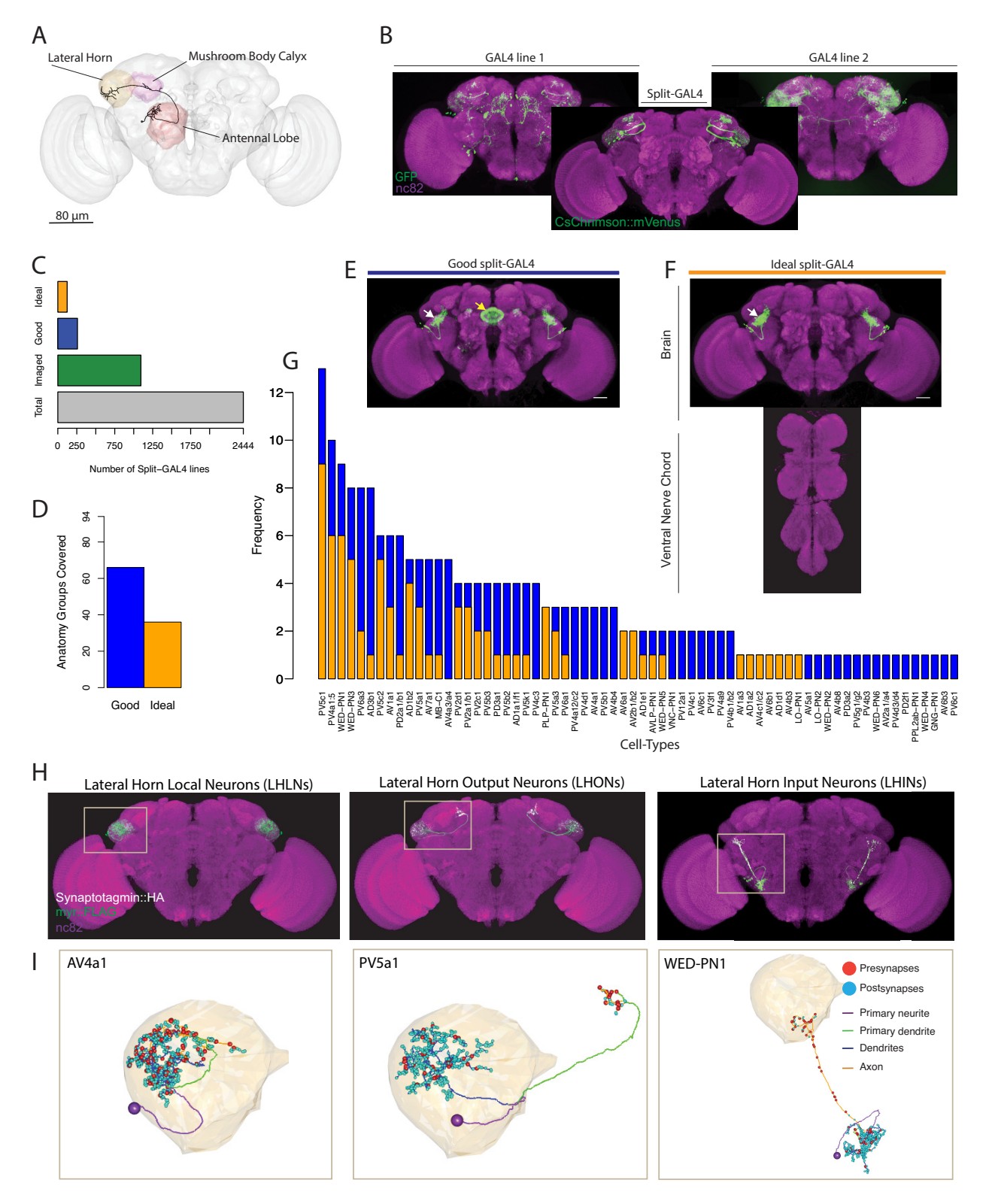

**Figure 1.** Generation and characterisation of a split-GAL4 lines for LH neurons. (**A**) A volume rendering of the adult *Drosophila* brain (grey) with the antennal lobe (red), mushroom body calyx (magenta) and lateral horn (yellow) neuropils labelled. A uniglomerular projection neuron (mPN1, black) from the DA1 glomerulus projects to the calyx and lateral horn. (**B**) Illustration of a split-GAL4 intersection. The expression patterns of two first-generation GAL4 lines are shown along with a split-GAL4 line (center) made by using the enhancer fragments from lines 1 and 2 to create GAL4 AD and DBD

*Figure 1 continued on next page*

*Figure 1 continued*

hemidrivers. Placing the two hemidrivers in the same fly results in a split-GAL4 line (foreground) specifically labelling an LH cell type that was present in both GAL4 lines. (C) Numbers of split-GAL4 combinations screened (grey), that underwent immunohistochemistry (green) and that were good (blue) and ideal (orange) out of the total 2444 split-GAL4 lines. See text and *Figure 1E–F* for definitions of good and ideal. (D) Number of known LH anatomy groups (as opposed to cell types, see text for definitions) targeted in this study covered by at least one good (blue) and ideal (orange) split-GAL4 line. (E) Example of a good split-GAL4 line, clearly labelling a LH cell type (white arrow) with off-target expression in the ellipsoid body (yellow arrow). (F) Example of an ideal split-GAL4 line, labelling the same cell type as E (white arrow). Note absence of off-target expression in brain. For E-F, scale bar represents 30 μm. (G) Number of good and ideal split-GAL4 lines identified across all LH cell types. (H) Segmented projections of example LH cell types for LHLNs (1D, left), LHONs (PV5a1, center) and LHINs (70C, right). A presynaptic marker (Syt::smHA, white) and a membrane marker (myr::smFLAG, green) is expressed in the neurons. The images are registered to the JFRC2013 template brain (magenta). (I) Reconstruction of a single neuron of the same cell types as (H) in the whole brain EM volume. Single neuron example of LHLN (AV4a1, left), LHON (3A, middle) and LHIN (WED-PN1, right). Polyadic presynapses are labelled with red spheres. Postsynapses are labelled with blue spheres. Dendritic and axonal domains are illustrated in different colours.

DOI: https://doi.org/10.7554/eLife.43079.002

The following figure supplement is available for figure 1:

**Figure supplement 1.** Further analysis, examples and characterization of split-GAL4 lines.

DOI: https://doi.org/10.7554/eLife.43079.003

driven aggression or courtship (*Clowney et al., 2015*; *Kohl et al., 2013*; *Kurtovic et al., 2007*; *Wang and Anderson, 2010*). One study in the *Drosophila* larva identified neurons that mediate olfactory induced feeding (*Wang et al., 2013*). However, the diversity, anatomy and individual function of LH cell types is scarcely known. A previous study classified two broad groups of LH neurons (*Fişek and Wilson, 2014*) but subsequent work identified several other cells that do not fall into these categories (*Aso et al., 2014b*; *Fişek and Wilson, 2014*; *Frechter et al., 2019*; *Jeanne et al., 2018*; *Liang et al., 2013*; *Strutz et al., 2014*), and the total number of cell types with substantial LH dendrite is estimated to be >165 (*Frechter et al., 2019*).

To better understand the circuits that mediate innate olfactory behaviour, in this study we developed a suite of cell type specific tools in *Drosophila* to gain access to, map and manipulate many of the lateral horn (LH) cell types. As described above, the major impediment for functional and anatomical analysis of the LH has been the lack of reagents specifically and sparsely labelling the constituent neurons. This issue is further compounded by the lack of conspicuous anatomical landmarks or structure in the LH. A previous neuroanatomical screen of more than 4000 enhancer trap lines identified only six with clear LH expression (*Tanaka et al., 2004*), a very low yield in comparison to similar work in the AL and MB (*Tanaka et al., 2008*; *Tanaka et al., 2012*). This suggests some property of LH neurons makes them difficult to target with traditional enhancer trap approaches. In this study we leverage a combination of enhancer-driven expression (*Pfeiffer et al., 2008*) and the intersectional split-GAL4 (*Luan et al., 2006*; *Pfeiffer et al., 2010*) system to produce strong and specific reagents to control many LH cell types, opening up this region to direct experimentation. To augment this resource, we classify the polarity and main neurotransmitter of 82 identified LH cell types, demarcating excitatory and inhibitory circuit components in this region. Using image registration we identify the superior lateral protocerebrum (SLP as the major output zone of the LH and the principal next node of higher olfactory processing. We also identify multisensory inputs from the visual, mechanosensory and gustatory systems that coproject with thermosensory inputs to a restricted ventral zone of the LH. To examine the links between innate and learned behaviour, we investigated potential interactions between the LH and MB, discovering many potential sites of convergence between the LH neurons, MBONs and DANs and found a new circuit motif where the axons of LH projection neurons integrate multiple MBON inputs. Finally, to better understand the function of LH neurons, we used targeted expression of an optogenetic effector to drive activity in populations of neurons. We identified LH cell types that drive attraction, aversion and specific motor behaviours when stimulated.

## Results

### Neurogenetic reagents for manipulating the Lateral Horn

In order to generate genetic driver lines that sparsely label LH neurons we used the split-GAL4 approach, an intersectional extension of the GAL4/UAS binary transcription factor system (*Luan et al., 2006*). In this approach, the GAL4 protein is divided into its two components, the activation domain and the DNA-binding domain, each of which is driven by a separate enhancer (*Luan et al., 2006*; *Pfeiffer et al., 2010*). When these two components are expressed in the same cell, the reconstituted GAL4 then drives transcription of the effector downstream of the UAS site, resulting in intersection of the two enhancer expression patterns. By combining enhancer fusions that share only a handful of cells, it is possible to create sparse and strong driver lines that specifically control cells or neurons of interest (*Figure 1B*). As several recent studies have developed specific split-GAL4 lines generated from sparse enhancer fusion GAL4 collections (*Aso et al., 2014a*; *Bidaye et al., 2014*; *Namiki et al., 2018*; *Tuthill et al., 2013*; *Wu et al., 2016*), we reasoned that this approach may circumvent the difficulty of manipulating LH neurons encountered in previous work.

We generated and screened 2444 split-GAL4 lines (*Figure 1C*) with mVenus-tagged csChrimson as a reporter (*Klapoetke et al., 2014*), based on a manual annotation of two large enhancer GAL4 collections (*Jenett et al., 2012*; *Kvon et al., 2014*) (see Materials and methods for full details). In an initial screening phase, we mounted and screened brains by native fluorescence only, which ensured a minimal strength of expression. Of these, 1095 split-GAL4 combinations containing both visible and sparse expression were selected for the second selection step, whole mount immunohistochemistry and confocal imaging (*Figure 1C*). In this stage, we identified two types of useful split-GAL4 lines. The first class had clear and unoccluded LH expression in a small group of anatomically similar neurons, but contained off-target expression elsewhere in the brain or ventral nerve cord (VNC). These were termed 'good', and could be used for a variety of experiments (*Figure 1E*). The second classification was 'ideal', these lines contained only one similar group of LH neurons and no additional expression in either the brain or VNC (*Figure 1F*, see *Figure 1—figure supplement 1A* for more examples of 'ideal' split-GAL4 brain and VNC). Of these 1095 split-GAL4 combinations that were stained and imaged, we identified 259 'good' and 123 'ideal' split-GAL4 lines. We selected a subset of these that specifically labeled one genetically identifiable group of LH neurons each and present 123 of these 'good' and 87 'ideal' split-GAL4 lines for use in the current study. In addition we identified, but did not further analyse lines that labeled more than one clearly identifiable group of LH neurons (see *Supplementary file 1–2*). Each unique combination of hemidrivers was given a line identifier, for example LH1989.

The LH is a complex neuropil consisting of ~1400 neurons with diverse morphologies but lacking anatomical landmarks to categorize neurons, such as the glomeruli of the antennal lobe (*Frechter et al., 2019*). To classify the different neurons identified in the LH, we used the hierarchical nomenclature developed in accompanying work (*Frechter et al., 2019*). This system uses three different features of increasing neuroanatomical detail to categorize neurons (*Figure 1—figure supplement 1B'*): primary neurite tract, anatomy group and cell type. The primary neurite tract is the region of the neuron separating the soma from the rest of the cell's axons and dendrites. In the fly brain, neurons of the same cell type always enter the neuropil via the same primary neurite tract. Frechter et al. identified 31 primary neurite tracts, named by their relationship with the LH, for example Posterior Ventral tract five or PV5 (see *Figure 1—figure supplement 1B"* for all primary neuron tracts identified). The primary neurite splits into an axonal and dendritic arborization (e.g. PV5a or PV5b, *Figure 1—figure supplement 1B'*), and we group neurons into classes with similar axon and dendrite projections, termed anatomy groups. Neurons of the same anatomy group project to similar sub-volumes of the brain and may overlap in space, but do not necessarily share exactly the same morphology. Distinct anatomical cell types (PV5b1 or PV5b2, *Figure 1—figure supplement 1B'*) were defined as neurons that have distinguishable axonal and/or dendritic arbours within an anatomy group, assisted by clustering of single-neuron morphology (*Costa et al., 2016*). Neurons of an anatomically-defined cell type are likely to receive similar dendritic input and supply similar axonic output (*Dolan et al., 2018*; *Gerhard et al., 2017*; *Jeanne et al., 2018*; *Tobin et al., 2017*), and LH neurons of an anatomical cell type respond similarly to different odour stimuli (*Frechter et al., 2019*). These anatomical cell types are likely the functional units of the LH.

As our annotation of GAL4 enhancers was based on full driver expression patterns rather than single-neuron data we targeted our split-GAL4 screen towards anatomy groups. Our annotations identified 93 genetically identifiable LH cell types present in available GAL4 collections. For 66 of these anatomical groups we identified at least one 'good' line while 36 had at least one 'ideal' split-GAL4 line (*Figure 1D*). Although we selected split-GAL4 lines based on anatomy groups, we next wanted to determine the cell type composition of these LH split-GAL4 lines in order to know precisely what neurons are being genetically manipulated. To do this we performed single neuron labelling (*Nern et al., 2015*) followed by morphological clustering (*Costa et al., 2016*) on split-GAL4 lines labelling different anatomy groups (see www.janelia.org/split-gal4 for 3D image data download of brain and VNC). In total our split-GAL4 lines covered 82 cell types of the LH (*Figures 2–5*, see *Supplementary file 1* for summary of all cell types and their anatomical details). Confocal image stacks of these neurons are available at splitgal4.janelia.org (Lateral Horn 2018). 3D skeletons of each anatomical cell type organized according to the hierarchical nomenclature system introduced in *Frechter et al., 2019* as well as their corresponding Split-GAL4 have been added to a web application at jefferislab.org/si/lhlibrary. This website also links cell types to LH cell type odor response data (*Frechter et al., 2019*) and functional connectivity data describing PN inputs (*Jeanne et al., 2018*).

Split-GAL4 lines of 53 anatomical groups matched to a single cell type. However, we found 12 anatomical groups isolated by split-GAL4 lines that appear to contain two cell types when single neuron morphologies were examined. As these pairs consistently appeared together and we could not separate them genetically with our split-GAL4 lines we grouped these cell type pairs together as a unit (e.g. PV2a1 and PV2b1 becomes PV2a1/b1) for further analysis. In addition, we identified one extreme case where five anatomically similar cell types (PV4a1:a5) of local neurons co-occurred in multiple driver lines (see below).

The distribution of 'good' and 'ideal' lines per cell type (rather than anatomy group) is plotted in *Figure 1G*. See *Supplementary file 1* for a list of the sparsest lines available per cell type and *Supplementary file 2* for all split-GAL4 lines identified, constituent cell types and hemidrivers. We identified either the same or likely matching cell types for LH neurons identified in previous studies (*Aso et al., 2014b*; *Dolan et al., 2018*; *Fişek and Wilson, 2014*; *Liang et al., 2013*; *Strutz et al., 2014*) with the exception of *fruitless*-positive neurons. We identified only one of the five *fruitless*-positive LH output neurons (*Cachero et al., 2010*), possibly because we screened only female animals for split-GAL4 expression. In summary, using an intersectional genetic approach, we developed genetic reagents for manipulating many cell types in this previously intractable higher olfactory neuropil.

## An anatomical resource for the lateral horn connecting light and electron microscopy

With these new genetic reagents we could gain insight into the anatomical organization of the *Drosophila* LH. To create a comprehensive functionally annotated atlas of identified LH neurons we determined the polarity (e.g. input, output and local neurons) and sign (inhibitory or excitatory) of each LH cell type using our split-GAL4 lines (see Materials and methods and www.janelia.org/split-gal4 for image data). We also sought to identify a matching neuron in a synapse-resolution electron microscopy (EM) dataset (*Zheng et al., 2018*) to provide a link between these different types of data and to enable future synaptic resolution connectomic studies. Note that in this study we use the terms *LH neuron* and *LH cell type* to refer to any individual neuron or identified cell type associated with the LH; this includes neurons providing input to the LH. In contrast, the acronym 'LHN' has previously been used to refer specifically to neurons with significant dendrites in the LH (*Fişek and Wilson, 2014*; *Frechter et al., 2019*; *Jeanne et al., 2018*).

To determine neuronal polarity we drove expression of a presynaptic marker (synaptotagmin-smGFP-HA) in each LH cell type and performed high-resolution confocal imaging to identify presynaptic sites (*Aso et al., 2014a*). We identified 44 LH output neuron (LHON), 8 LH local neuron (LHLN) and 14 non-antennal lobe input neuron (LHIN) cell types (*Figures 2–5*, see *Figure 1H* for an example cell type of each polarity category). We rarely observed complete polarisation in LHONs, many of which had some presynaptic labelling in their dendrites (see *Figure 1—figure supplement 1C* for illustrative example of cell type PV5a1). This is similar to mPN1s which have been shown to form dendrodendritic synapses (*Kazama and Wilson, 2009*; *Rybak et al., 2016*). For one likely

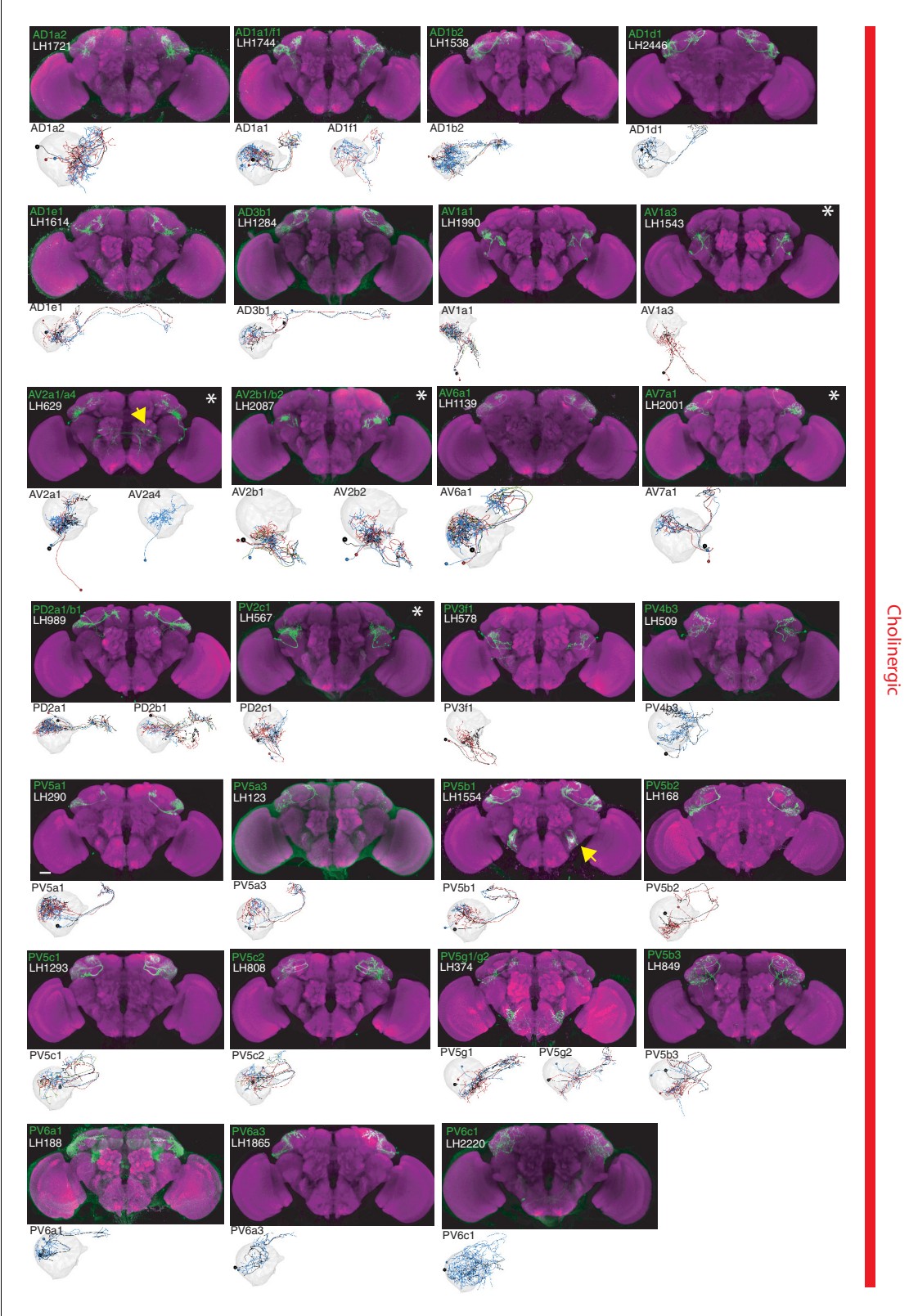

**Figure 2.** The most specific split-GAL4 lines for identified cholinergic LHONs. Array of projections of the most specific split-GAL4 line for each cholinergic LHON cell-type. Where available, a neuron from that class traced to at least to identification in the EM volume (inset, black represents tracing, grey the lateral horn, blue and brown the single neuron labelling, **Nern et al., 2015**). Expression of split-GAL4 lines are visualized using UAS-csChrimson::mVenus in *attP18* (green), with nc82 as a neuropil stain (magenta). Each image is registered to JFRC2010, while the EM data is registered

*Figure 2 continued on next page*

*Figure 2 continued*

to JFRC2013. The cell-type (eg. PV5a1) labelled is in the top left of each panel, while the line code (eg. LH290) is listed below. Cell-types that stained positive for more than one neurotransmitter are labelled with an asterix in the top right. Off-target expression in the brain for non-ideal lines labelled with a yellow arrow. See www.janelia.org/split-gal4 for image data.

DOI: https://doi.org/10.7554/eLife.43079.004

The following figure supplement is available for figure 2:

**Figure supplement 1.** Examples of cell type staining positive for two neurotransmitters, examples of bilateral LHONs.

DOI: https://doi.org/10.7554/eLife.43079.005

dopaminergic cell type isolated we could not determine polarity and did not further analyse these neurons (*Figure 1—figure supplement 1D*).

To validate our polarity categories, we leveraged a whole brain synaptic-resolution electron microscopy (EM) volume (*Zheng et al., 2018*). For the three example cell types in *Figure 1H*, we identified and comprehensively reconstructed a neuron of that same cell type in the EM dataset (*Figure 1I*). *Drosophila* synapses are polyadic, with each presynapse (red spheres in *Figure 1I*) having multiple postsynaptic targets. By labelling both their pre- and postsynapses (*Figure 1I*), we classified these neurons into the same polarity categories, supporting our light-microscopy analysis of neuronal polarity. Interestingly, for both our exemplar LHON (PV5a1) and LHIN (WED-PN1) we found incomplete polarization, suggesting that information flow can occur in both directions (*Figure 1I*).

To understand information flow to and from the LH, it is essential to know if a given cell type is excitatory or inhibitory. To identify principal fast neurotransmitters, we performed immunohistochemistry for each of the three main neurotransmitters in the fly brain (glutamate, acetylcholine and GABA) (*Figure 1—figure supplement 1E'–E''*). In *Drosophila*, the major excitatory and inhibitory neurotransmitters are acetylcholine and GABA respectively, while both excitatory and inhibitory ionotropic glutamate receptors are expressed in the brain (*Liu and Wilson, 2013*). The majority of identified LH cell types stained positive for one of these neurotransmitters (*Figures 2–5*), although we did not determine the presence of neuropeptides (such as neuropeptide F) or monoamines (such

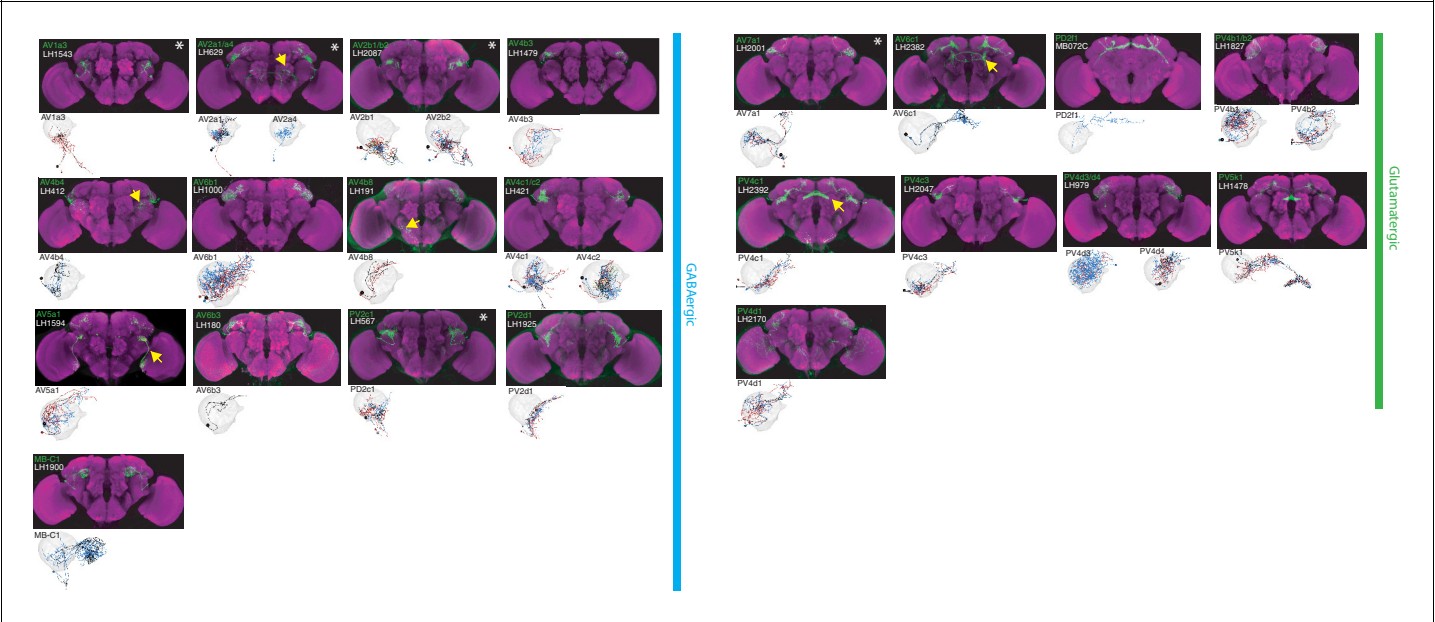

**Figure 3.** The most specific split-GAL4 lines for identified GABAergic and glutamatergic LHONs. Array of projections of the most specific split-GAL4 line for each LHON cell-type. All LHON cell-types are organized by principle fast neurotransmitter: GABAergic and Glutamatergic. Images are otherwise presented as Figure 2. Note that MB-C1 was identified and named in a previous study (*Aso et al., 2014a*). See www.janelia.org/split-gal4 for image data.

DOI: https://doi.org/10.7554/eLife.43079.006

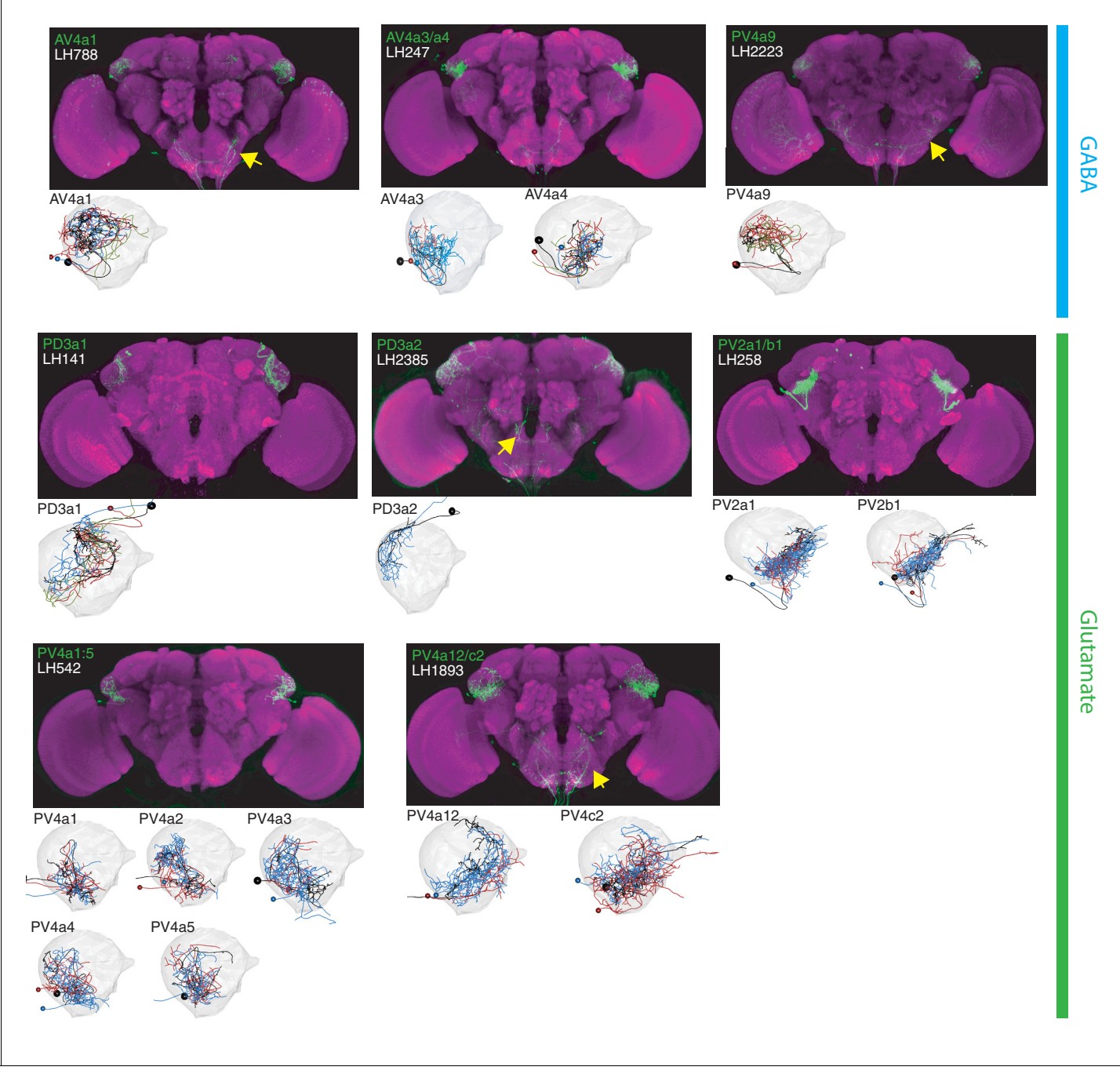

**Figure 4.** The most specific split-GAL4 lines for identified LHLNs. Array of projections of the best split-GAL4 line for each LHLN and where available, a neuron from that class traced in the EM volume (inset, black represents tracing, red the lateral horn, blue and brown the single neuron labelling, *Nern et al., 2015*). Each split-GAL4 line is expressing csChrimson::mVenus in *attP18*, and has nc82 as neuropil stain (magenta). Each image is registered to JFRC2010, while the EM data is registered to JFRC2013. The cell-type labelled is in the top left of each panel, while the panel number and line code is listed on the bottom left (eg. LH542). All cell-types are organized by fast neurotransmitter (GABA or glutamate). Cell-types that stained positive for more than one neurotransmitter are labelled with an asterix in the top right. See www.janelia.org/split-gal4 for image data.

DOI: https://doi.org/10.7554/eLife.43079.007

The following figure supplement is available for figure 4:

**Figure supplement 1.** LHLN tracts, novel PNs and an identified ascending LHIN.

DOI: https://doi.org/10.7554/eLife.43079.008

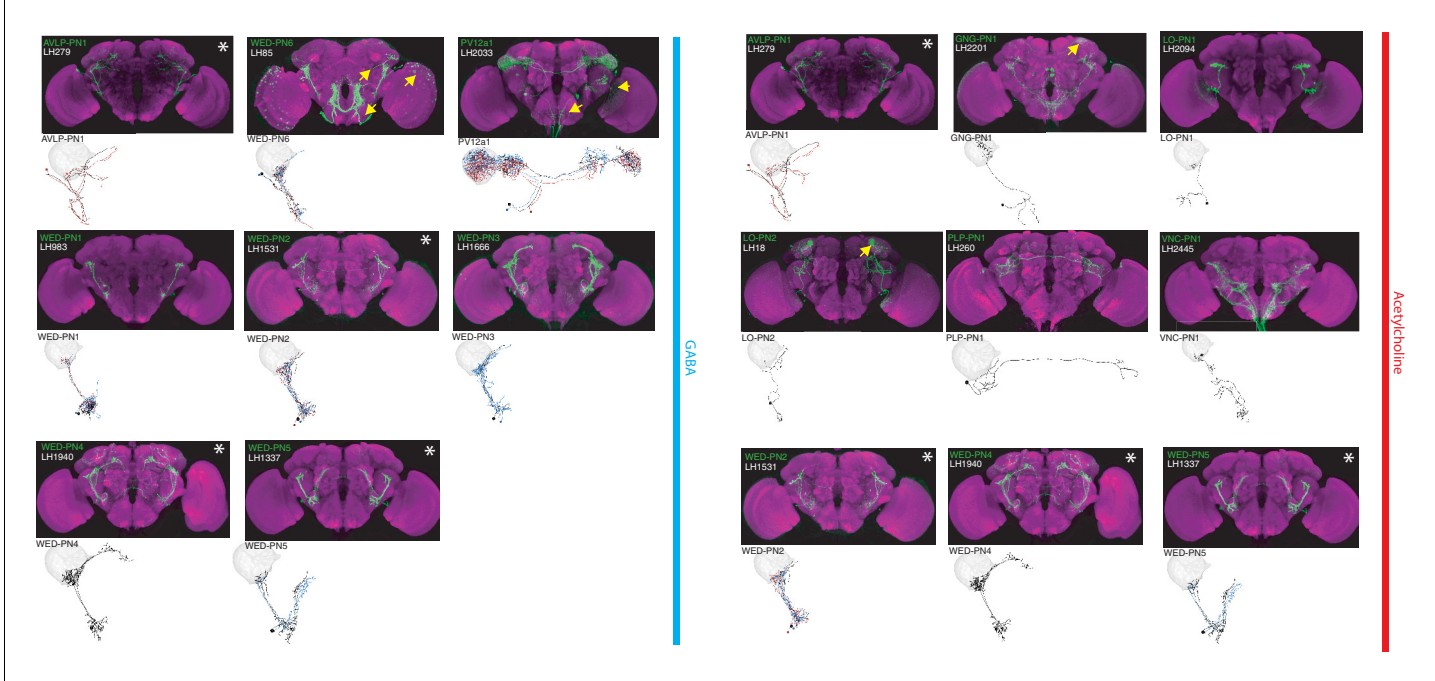

**Figure 5.** The most specific split-GAL4 lines for identified LHINs. Array of projections of the best split-GAL4 line for each LHIN cell-type and where available, a neuron from that class traced in the EM volume (inset, black represents tracing, red the lateral horn, blue and brown the single neuron labelling, *Nern et al., 2015*). Each split-GAL4 line is expressing csChrimson::mVenus in *attP18*, and has nc82 as neuropil stain (magenta). Each image is registered to JFRC2010, while the EM data is registered to JFRC2013. The cell-type labelled is in the top left of each panel, while the panel number and line code is listed on the bottom left. All cell-types are organized by principle fast neurotransmitter (GABA or acetylcholine). Cell-types that stained positive for more than one neurotransmitter are labelled with an asterix in the top right. For LHINs, we named the majority according to their main region of dendritic arborization using the shorthand defined in *Ito et al. (2014)*. See www.janelia.org/split-gal4 for image data.
DOI: https://doi.org/10.7554/eLife.43079.009

as octopamine) in these neurons. For cell types that were GABA-positive, in all cases the signal was evident across all neurons in the tested split-GAL4 line. However, we did not have the resolution to determine if all boutons in a line were cholinergic or glutamatergic (see Materials and methods).

The availability of whole brain EM mapping is a major advantage of studying sensory processing in *Drosophila* (*Eichler et al., 2017*; *Ohyama et al., 2015*; *Zheng et al., 2018*). We sought to provide a link between genetically controlled LH neurons and this synapse-resolution data by tracing and identifying backbones of neurons for identified LH cell types. We used our single neuron data and image registration to match split-GAL4 lines with most probable EM skeletons. For the majority of cell types described in this study, we identified and traced a backbone in the EM volume (*Zheng et al., 2018*) of a neuron likely from that cell type (*Figures 2–5*, insets) (See online Methods for data dissemination). Although most of these cells are not traced to completion, they provide a resource that will enable investigators to identify their pre- and postsynaptic partners. Combined with our split-GAL4 driver lines, polarity data and neurotransmitter stainings, this provides a resource to both functionally manipulate and exhaustively map pre- and postsynaptic connectivity of identified LHONs.

## Lateral Horn Output Neurons (LHONs)

By driving expression of a presynaptic marker in LH neurons, we identified 44 LHON cell types (or pairs of cell types) in our split-GAL4 collection. In *Figures 2* and *3* we present the most specific split-GAL4 lines ('ideal' lines where possible) for LHON cell types, sorted by principal neurotransmitter (*Figures 2* and *3*). Of these LHON cell types, 26 were acetylcholinergic (*Figure 2*), 13 GABAergic (*Figure 3*) and nine were glutamatergic (*Figure 3*), indicating both excitatory and inhibitory information flow from the LH to its target neuropils. Unexpectedly, we identified three cell types expressing multiple fast neurotransmitters (*Figures 2* and *3*, white asterisks). Two cell types, AV2a1/a4 and

AV2b1/b2, were identified as both cholinergic and GABAergic, indicating dual excitatory and inhibitory action. Similarly, AV7a1 expressed markers for both acetylcholine and glutamate (*Figure 2—figure supplement 1A*). Neurons with similar dual neurotransmitter profiles in the *Drosophila* brain have been independently identified by single-cell transcriptomics (*Croset et al., 2018*).

Identified LHONs projected to a variety of different brain regions (*Figures 2* and *3*). As the presumptive site of innate olfactory processing, we were surprised to identify no direct descending output to the ventral nerve cord, although this was consistent with previous studies (*Hsu and Bhandawat, 2016*; *Namiki et al., 2018*). We found only one cell type projecting to the central complex, a sensory-motor integration center (*Pfeiffer and Homberg, 2014*; *Wolff et al., 2015*) (glutamatergic LHON PV5k1 projecting to layer 2 of the fan-shaped body) and no direct projections to the ellipsoid body, which has been implicated in aversive chemotaxis (*Gao et al., 2013*). This indicates at least one more layer of sensory processing must exist before olfactory information is sent to the ventral nerve cord to elicit motor behaviour.

We also identified three bilateral LHONs connecting both hemispheres of the brain and confirmed their projections using single-cell labelling (*Nern et al., 2015*). Two were cholinergic bilateral LHONs, AD1e1 a *fruitless*-positive cell type also known as aSP-g (*Cachero et al., 2010*; *Kohl et al., 2013*) and AD3b1, while the other was PV5k1 (*Figure 2—figure supplement 1B*). These neurons have dendrites in one LH and project to both the ipsilateral and contralateral output zones, providing a possible mechanism for how the higher olfactory system communicates with both hemispheres of the brain to produce coordinated behaviour.

## Lateral Horn local neurons (LHLNs)

Our polarity analysis confirmed the local nature of 8 LHLN cell types or groups of cell types, for which we had developed split-GAL4 lines. *Figure 4* displays the sparsest such lines sorted by principal neurotransmitter. Local neuron axons typically covered subdivisions of the LH, with several cell types restricted ventrally (e.g. PV2a1/b1, PV4a12/c2), others more dorsally (e.g. AV4a1, PD3a2) and one cell type restricted to the medial peripheral LH (PD3a1). LHLNs were either GABAergic or glutamatergic (*Figure 4*). Indeed we found the majority of local neuron cell types were constituents of four different primary neurite tracts, with all cell types in a tract positive for the same neurotransmitter (glutamatergic PV4 and PD3, GABAergic AV4 and PV2) (*Figure 4—figure supplement 1A*). These data indicate that while lateral inhibition certainly exists in the LH (*Fişek and Wilson, 2014*), there is also the potential for lateral excitation via excitatory glutamatergic signalling (*Miyashita et al., 2012*).

## Lateral Horn Input Neurons (LHINs)

The final polarity category we identified was LHINs, providing input to the lateral horn. As we did not target olfactory PNs in our split-GAL4 screen, we focus here mostly on LHINs from brain regions other than the antennal lobe (However, we did identify split-GAL4 lines labelling two non-canonical PN cell types, see *Figure 4—figure supplement 1B–C*). We identified 'good' or 'ideal' lines for 14 LHINs and characterized their principal fast neurotransmitter (*Figure 5*), finding that LHINs were either cholinergic, GABAergic or both.

We found a large number of LHIN cell types projecting from regions other than the AL. These Novel LHINs include projections from the wedge and inferior posterior slope in the auditory and mechanosensory system (*Figure 5*, WED-PN1-6) (*Clemens et al., 2015*; *Matsuo et al., 2016*), the lobula in the visual system (*Figure 5*, LO-PN1-2) and the gustatory subesophageal zone (*Figure 5*, GNG-PN1). We also identified a putative ascending neuron with dendrites in the gustatory system (*Figure 5*, VNC-PN1 and *Figure 4-figure supplement 1D'-D''*). Although both split-GAl4 lines labelling VNC-PN1 were merely good, (*Figure 4—figure supplement 1D'*) cell bodies were solely present in the ventral nerve cord and the neurons drove a strong presynaptic marker signal in the ventral LH (*Figure 4—figure supplement 1D''*) strongly suggesting this neuron sends input from the VNC to the LH, potentially providing mechanosensory or pheromonal information (*Ramdya et al., 2015*; *Thistle et al., 2012*).

## Anatomical organization of LH local and output neurons

We next wanted to systematically compare the axons and dendrites of LH neurons to identify putative circuit motifs that may mediate innate olfactory processing. We generated 3D masks of the neurites of LHONs, LHINs and LHLNs (*Figure 6—figure supplement 1A*) described in *Figure 2–5* registered to the JFRC2013 template brain (*Aso et al., 2014a*; *Jefferis et al., 2007*; *Rohlfing and Maurer, 2003*). For LHONs and LHINs, we used the polarity signal to guide our division of the cell types into the predominantly axonal or dendritic compartments. By comparing these masks in 3D, we could calculate an overlap score which can be used as a proxy for proximity, shared downstream targets and potential synaptic connectivity (*Dolan et al., 2018*)(see Discussion for advantages and limitations of this approach).

To determine higher olfactory processing centers we calculated the overlap score between LHON axons and identified neuropils in the fly brain (*Figure 6—figure supplement 1B*) (*Ito et al., 2014*). Although LHONs had axons in 24 different neuropils, we found the majority of LH output converged to the superior lateral protocerebrum (SLP), where arborization was broad (*Figure 6—figure supplement 1C*), followed by two other nearby neuropils, the superior intermediate protocerebrum (SIP) and the superior medial protocerebrum (SMP) (*Figure 6—figure supplement 1A* and *Figure 6A*). This indicates that these regions, in particular the SLP are the next, third-order olfactory processing nodes.

We next examined the downstream convergence of LH information by calculating an overlap score between LHON axons (*Figure 6B*). We found two clear clusters of LHONs with axons co-projecting ot the same location, both in the SLP (*Figure 6B*, orange and blue borders). One of these received mostly excitatory LHONs (*Figure 6B*, blue) while the other cluster received a mix of cholinergic, GABAergic and glutamatergic LHONs (*Figure 6B*, orange), possibly bidirectionally modulating common downstream neurons. Given their axonal proximity, these LHONs likely share downstream targets. We overlaid two example cell types from each cluster in the SLP, revealing that this region can be subdivided into at least two distinct domains based on information flow from the LH (*Figure 6C*, orange and blue dashed circle). We did not find a strong relationship between LHON axonal and dendritic overlap ($R^2$ = 0.094, *Figure 6—figure supplement 1D*).

We next used this overlap analysis approach to estimate how populations of LHLNs (*Figures 4* and *5*) interact with the dendrites of LHONs, to determine if there is an internal structure in the LH (*Figure 6D*). We found that LHLNs could be sorted into two broad groupings based on the degree of overlap (most or many) with LHON dendrites in the lateral horn (*Figure 6D*). LHLNs ramified across broad regions of the LH, with each LHLN exhibiting potential interactions between several LHONs (e.g. AV4a1 in *Figure 6E*). As LHONs are not completely polarized however, it is not possible to predict the direction of these interactions. This is similar to the AL, where PNs both transmit and receive input from local neurons (*Liu and Wilson, 2013*; *Yaksi and Wilson, 2010*).

## Mapping excitatory, uniglomerular Projection Neuron input to the LH

As the LH is downstream of the AL, could we use the above overlap analysis to compare PN arborization with the dendrites of different LH neuron cell types? As there are few genetic tools available to specifically control PN cell types, we relied on published data registered onto the template brain used in this study (JFRC2013) by bridging multiple registrations (*Manton et al., 2014*), although this increases variability due to additive registration error. We focus here on the excitatory, uniglomerular PNs (mPN1s) for which registrable single-neuron data has been published for a majority of glomeruli (*Costa et al., 2016*; *Grosjean et al., 2011*; *Jefferis et al., 2007*; *Lin et al., 2013*; *Yu et al., 2010*).

We first compared the axons of PNs with LHON dendrites using light microscopy and image registration. We found a broad range of overlap (*Figure 7—figure supplement 1A*). PN axons coarsely segregated LHON dendrites into three clusters, two groups of LHONs that potentially receive input from many PNs (*Figure 7—figure supplement 1A', e.*g. AV1a1, AV6b1) and one group of LHONs that overlap with the PNs of only a few glomeruli (*Figure 7—figure supplement 1A', e.*g. PV5k1, AV2b1/b2). We examined glomeruli that have clear functions in olfactory behaviour such as avoidance and approach to ecologically critical odours. The DA2, DL4 and VA7l glomeruli all specifically encode aversive olfactory odours (*Ebrahim et al., 2015*; *Mansourian et al., 2016*; *Stensmyr et al., 2012*) and potentially interact with a large number of LHONs (*Figure 7—figure supplement 1A',*

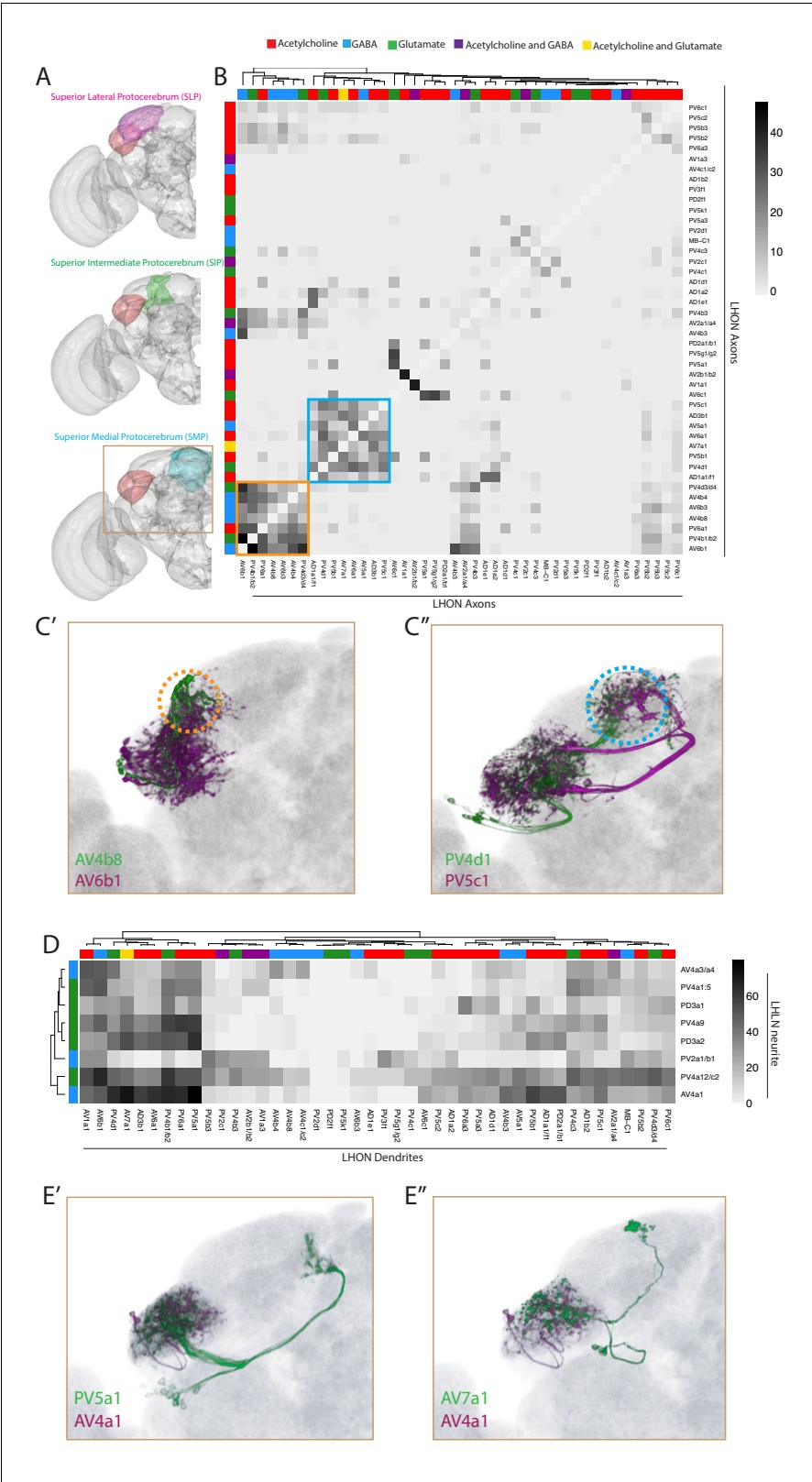

**Figure 6.** Inter- and intra-LH interactions. (**A**) Volume renderings of fly brain (grey) with the SLP (magenta), SIP (green) and SMP (cyan) labelled. The LH in each panel is labelled in red. The brown inset is the region displayed in C and E. (**B**) Heatmap of percentage overlap score (white to black) between masks of LHON axons against themselves. Scores between the same LHON axon and itself are set from 100 to −1 for clarity. The tracks of the rows and columns both represent the neurotransmitter, as determined by immunohistochemistry for acetylcholine, GABA and glutamate. Two main clusters of

*Figure 6 continued on next page*

*Figure 6 continued*
axons emerged from the analysis (orange and blue bound). (C') Volume rendering of AV4b8 (green) and AV6b1 (magenta), both are expression patterns from different brains registered to the JFRC2013 template brain. The orange dashed circle shows the axonal projections of these two cell types which coclustered in the heatmap orange box. (C'') Volume rendering of PV4d1 (green) and PV5c1 (magenta), both are expression patterns from different brains registered to the JFRC2013 template brain. The blue dashed circle shows the axonal projections of these two cell types which coclustered in the heatmap blue box. (D) Heatmap of percentage overlap score (white to black) between masks of LHON dendrites against the neurites of LHLNs. The tracks of the rows and columns both represent the neurotransmitter, for LHLNs and LHONs respectively (see *Figure 6B* for neurotransmitter colour code). (E') Volume rendering of PV5a1 LHON (green) and AV4a1 LHLN (magenta), both are expression patterns from different brains registered to the JFRC2013 template brain. (E'') Volume rendering of AV7a1 LHON (green) and AV4a1 LHLN (magenta), both are expression patterns from different brains registered to the JFRC2013 template brain.
DOI: https://doi.org/10.7554/eLife.43079.010
The following figure supplement is available for figure 6:

**Figure supplement 1.** Analysis of light microscopy data, LHON neuropil projections, LHON and LHLN interactions.
DOI: https://doi.org/10.7554/eLife.43079.011

magenta boxes), including AV6b1 and AV1a1) which have high overlap scores for all three glomeruli indicating possible convergence. Surprisingly, the VM1 glomerulus, which encodes attractive amines (*Min et al., 2013*) also had a high overlap score for these LHONs (*Figure 7—figure supplement 1A'*, green box). This suggests there may be convergence of attractive and aversive odour information onto several LHONs, although EM reconstruction will be needed to accurately determine connectivity.

PNs with dendrites in the V glomerulus mediate responses to $CO_2$ (*Lin et al., 2013*). Unlike most glomeruli (*Jefferis et al., 2007*; *Kazama and Wilson, 2009*), PNs from the V glomerulus have diverse anatomies and project to the LH in parallel to mediate $CO_2$ aversion in a concentration dependent manner (*Lin et al., 2013*). We categorized several of these neurons from the Flycircuit database (*Lin et al., 2013*) (see Materials and methods). We identified several LHONs with sparse PN overlap, several of which verlapped V glomerulus PNs. For example, PV5g1/g2 overlapped strongly with PN-$V_6$ (*Figure 7—figure supplement 1A''*). This analysis provides tools and potential downstream circuits for further studies of the neurons that mediate responses to $CO_2$, an ecologically critical odour for the fly (*Bräcker et al., 2013*; *Faucher et al., 2013*; *Lewis et al., 2015*). Finally, we compared the overlap between PNs and LHLNs, finding extensive overlap between the axons of PNs and LHLNs (*Figure 7—figure supplement 1B'*). We also identified PV2a1/b1 as a cell type that potentially receives $CO_2$ input (*Figure 7—figure supplement 1B''*), a finding that we have also confirmed electrophysiologically (*Frechter et al., 2019*).

## The Lateral Horn receives multimodal input in a restricted ventral zone

The LH is generally considered to be an olfactory processing center (*Masse et al., 2009*; *Schultzhaus et al., 2017*). However, in addition to olfactory input from the AL, we found several LHINs that projected from a variety of neuropils (see *Figure 7A*) including visual, mechanosensory and gustatory centers (*Figure 5*). Previous studies have also documented non-canonical inputs to the LH, projecting from both the thermosensory and taste systems (*Frank et al., 2015*; *Kim et al., 2015*). Moreover, cells recorded from the locust LH responded to both olfactory and visual stimuli (*Gupta and Stopfer, 2012*). As no study has looked at multimodal inputs together with LHLNs and LHONs, we wanted to integrate this novel input with our atlas of genetically identifiable neurons to determine how non-olfactory input relates to other neurons of the LH.

We assigned LHINs to a sensory modality if their dendrites ramified in neuropils associated with visual, gustatory or temperature processing (*Figure 7B*, column tracks, see Materials and methods). We also imaged and registered previously published GAL4 lines labelling four thermosensory and two taste LH-projecting cell types (*Frank et al., 2015*; *Kim et al., 2015*), segmenting the cells for the LHIN dataset (*Figure 7—figure supplement 1C*). To systematically compare non-olfactory input to the LH, we calculated the overlap between LHINs and LHONs (*Figure 7B*). We found potential interactions between LHON dendrites and neurons from all of the above modalities. For example the sole cell type projecting to the central complex (PV5k1) potentially receives cholinergic or GABAergic input from mechanosensory or wind-sensing LHINs (WED-PN3, *Figure 7C'*). We also identified LHONs that may receive visual input (for example AV2b1/b2, *Figure 7C''*) and

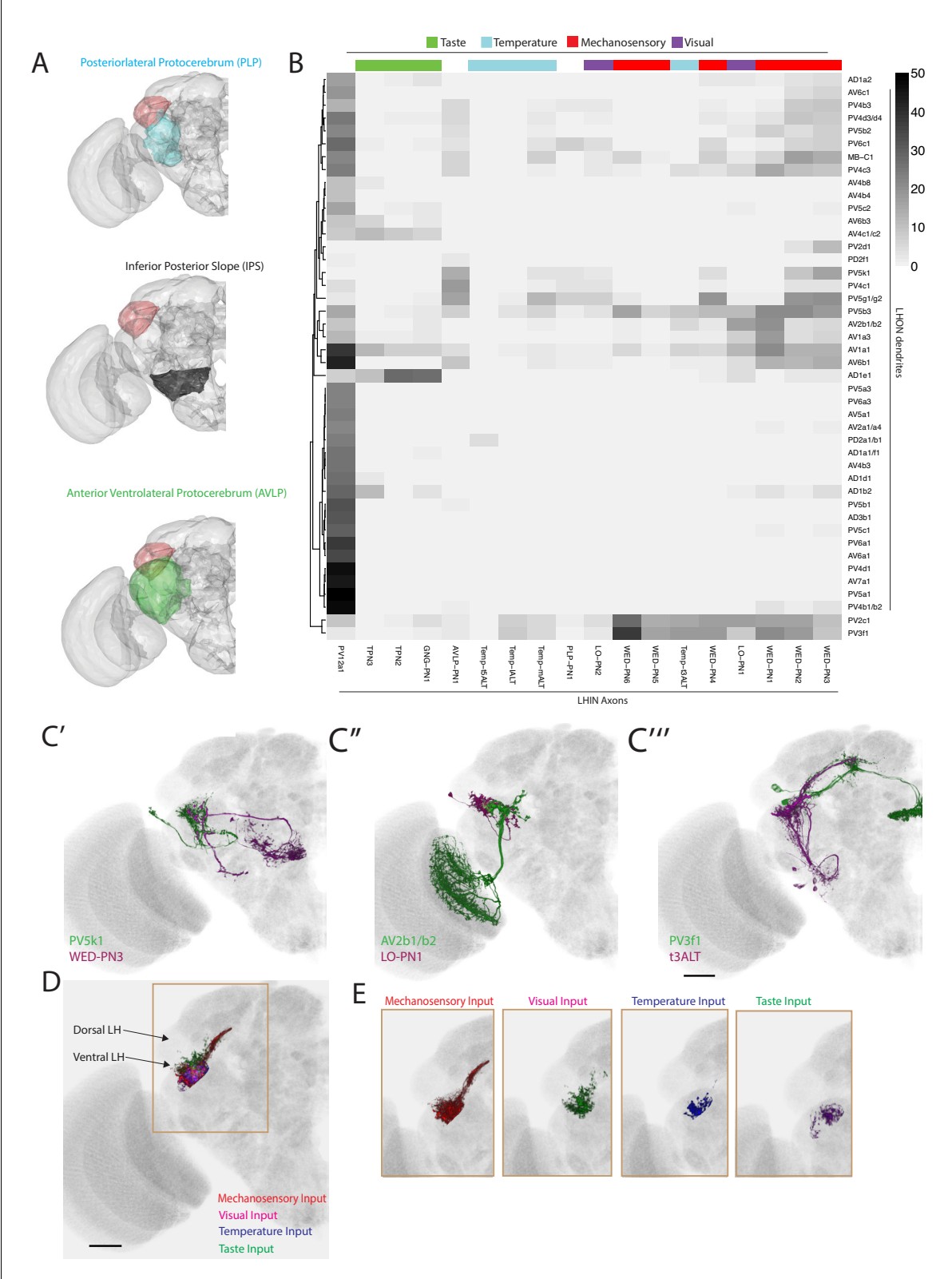

**Figure 7.** The LH receives non-olfactory input to a restricted ventral zone. (**A**) Volume renderings of fly brain (grey) with PLP (blue), IPS (black) and AVLP (green) labelled. The LH in each panel is labelled in red. (**B**) Heatmap of percentage overlap score between masks of LHON dendrites against the axons of non-canonical LHINs. The tracks of columns represent the presumed modality of the LHIN, as determined by position of their dendrites. Temp-ALT etc represents four different types of temperature PNs (*Frank et al., 2015*). (**C'**) Volume rendering showing overlap of PV5k1 (green) and WED-PN3

*Figure 7 continued on next page*

*Figure 7 continued*
(magenta), both of which are expression patterns from different brains registered to the JFRC2013 template brain. (**C''**) Volume rendering showing overlap of AV2b1/b2 (green) and LO-PN1 (magenta), both of which are expression patterns from different brains registered to the JFRC2013 template brain. (**C'''**) Volume rendering showing overlap of PV3f1 (green) and the temperature projection neuron t3ALT (magenta), both of which are expression patterns from different brains registered to the JFRC2013 template brain. (**D**) Mutimodal input to the LH is restricted to a ventral zone. Averaged axonal representations across different cell types for each non-olfactory sensory modality (mechanosensory, temperature, visual and taste) overlaid onto the JFRC2013 template brain demonstrating these inputs are restricted to the ventral LH. Scale bar is 30 μm. (**E**) Individual volume renderings of the averaged axonal representations (only pixels in the LH are displayed) for each sensory modality, visual (top left), temperature (top right), mechanosensory (bottom left) and taste (bottom right).
DOI: https://doi.org/10.7554/eLife.43079.012

The following figure supplement is available for figure 7:

**Figure supplement 1.** Input to the LH from PNs, MBONs and LHINs.
DOI: https://doi.org/10.7554/eLife.43079.013

temperature input (for example PV3f1, *Figure 7C'''*). These data can be combined with our PN input data to identify LHONs that may integrate multiple sensory modalities. For example, AV2b1/b2 was positioned to integrate both visual and olfactory information from several glomeruli such as VA4, potentially to add visual context to olfactory stimuli.

In addition to input from different sensory systems, the LH also receives experience-dependent input from the MB via several MBONs, namely MBON-α2sc (also known as MB-V2α), MBON-α′3 (also known as MB-V2α′) and MBON-α3 (also known as MB-V3) (*Aso et al., 2014a*; *Séjourné et al., 2011*; *Tanaka et al., 2008*). We calculated overlap between the axons of MBONs and the dendrites of LHONs, finding several instances of high overlap score (*Figure 7—figure supplement 1D'*). Several LHONs also had dendrites protruding outside the LH (e.g. AD1b2) and these dendrites overlapped with non-LH projecting MBONs, most obviously for AD1b2 and MBON-α1. We confirmed MBON-α1 synaptic input to AD1b2 in EM by fully reconstructing AD1b2 neurons (see below). In addition, we observed overlap between PD2a1/b1 and MBON-α2sc, for which connectivity has been validated with EM and functional imaging (*Dolan et al., 2018*). Within the LH, one striking observation was potential overlap between MBON-α′3 and the PV3f1 (*Figure 7—figure supplement 1D''*), suggesting that this cell type may integrate temperature and odour novelty (*Hattori et al., 2017*).

We observed that many of the non-olfactory modalities converged to the ventral region of the LH. To compare different input modalities we calculated averaged representations across cell types of each modality and selected only the pixels within a LH mask. This analysis clearly demonstrates that multimodal input to the LH is restricted to a limited ventral zone (*Figure 7D–E*), with the most extensive inputs coming from the mechanosensory and auditory system (*Figure 7E*). Thus, the LH consists of a multimodal ventral zone and an olfactory privileged dorsal segment with LHONs potentially capable of integrating multiple sensory modalities.

## Interactions between Lateral Horn Output Neurons and Mushroom Body neurons

The majority of identified LHONs project to the SLP and nearby neuropils, the same regions that are innervated by MB-associated neurons (*Aso et al., 2014a*). Could hardwired olfactory information interact with neurons that mediate the acquisition and retrieval of memories? While previous work identified three instances of potential convergence of LHONs with MBONs and/or DANs (*Aso et al., 2014b*; *Dolan et al., 2018*), it was unclear if this was a general property of all LHONs or a minor subset. Using the genetic tools developed here, we conducted a more comprehensive analysis across many LH cell types.

First we asked if LHONs axonal terminals were in close proximity to DAN dendrites, raising the possibility that they might directly impact dopaminergic modulation of Kenyon cells and thus the neuroplasticity of MB olfactory responses (*Cohn et al., 2015*; *Hige et al., 2015a*). We created masks for DAN dendrites and systematically compared their overlap with LHON axons (*Figure 8A*, see *Figure 8C'* for example overlay), finding that many, but not all LHONs converged with DAN dendrites. The proximity of these axons and dendrites suggests that many LHONs could be presynaptic to one or more DANs; if this is the case, then DANs may integrate input from several LHONs. LHONs were proximal to the dendrites of DANs from both the PAM and PPL1 clusters (*Mao and*

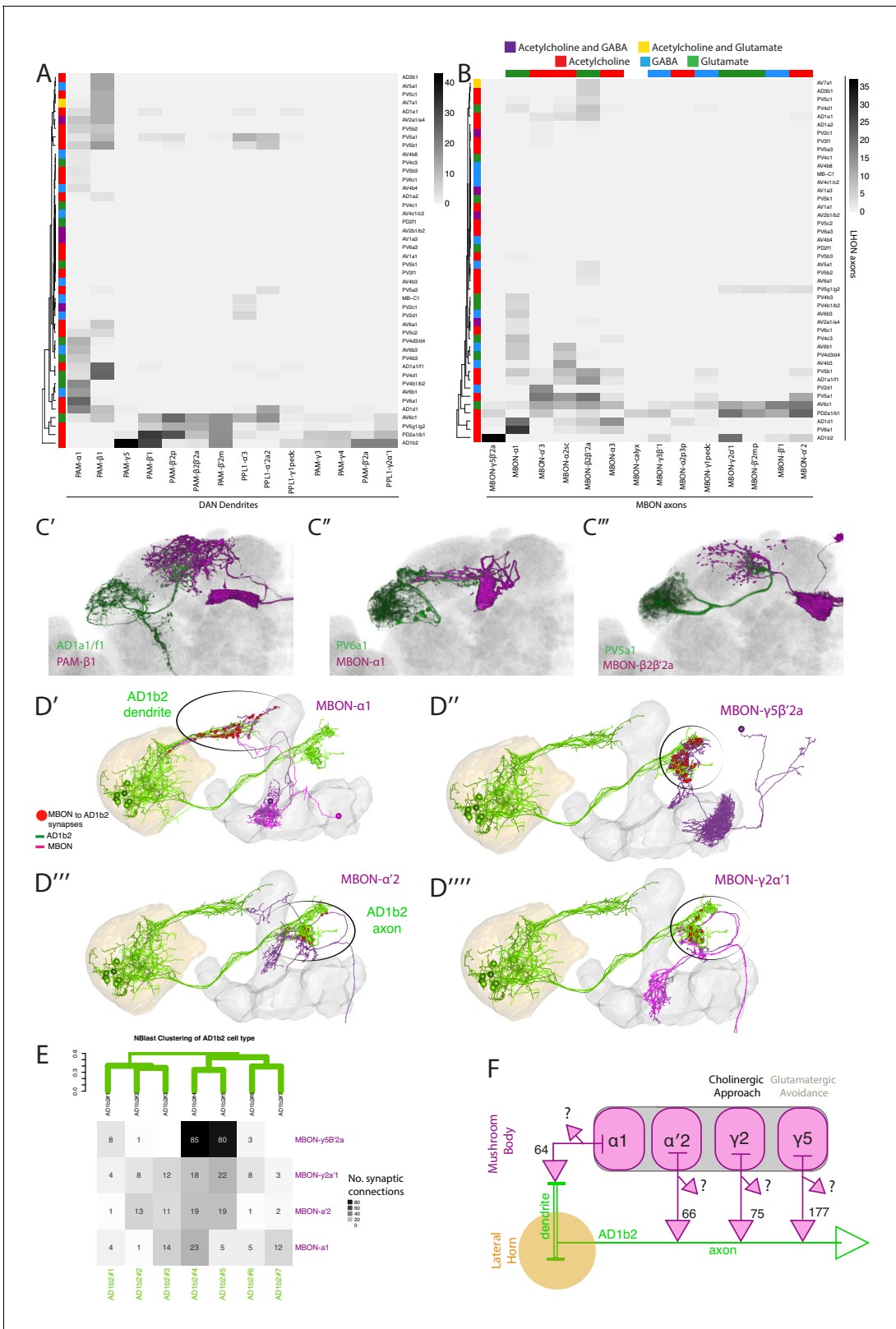

**Figure 8.** Identifying interactions between the innate and learned olfactory processing centers. (**A**) Heatmap of percentage overlap score between masks of LHON axons and DAN dendrites. The tracks of the rows represent the neurotransmitter, as determined by immunohistochemistry for acetylcholine, GABA and glutamate. (**B**) Heatmap of percentage overlap score between masks of LHON axons and MBON axons. The tracks of the rows and columns represent the neurotransmitter for LHONs and MBONs respectively. White asterisks represent cell type pairs that are illustrated with

*Figure 8 continued on next page*

*Figure 8 continued*

volume renderings (see below). (**C'**) Volume rendering showing overlap of AD1a1/f1 (green) and PAM-β1 (magenta) (**C''**) Overlap between PV6a1 and MBON-α1 (**C'''**) PV5a1 and MBON-β2β'2a. All of C'-C''' are expression patterns from different brains registered to the JFRC2013 template brain. (**D**) EM reconstructions of AD1b2 (green) and MBONs (magenta), illustrating synaptic connectivity from MBON-α1, MBON-γ5β'2a, MBON-α'two and MBON-γ 2α'one onto AD1b2. (**D'**) MBON-α1 forms synapses on AD1b2 dendrites which are outside of the LH. (**D''–D'''**) MBON-γ5β'2a, MBON-α'two and MBON-γ2α form axoaxonic synapses onto AD1b2 in the SMP. The LH and MB volumes are labelled in orange and light grey respectively. Red spheres represent synapses from MBON onto AD1b2. Black circle highlights region of synapses. (**D''**) Reconstruction of seven AD1b2 neurons (green) and MBON-γ2α1 (black) in the EM volume. The LH and MB volumes are labelled in red and light grey respectively. Red spheres represent synapses from MBON-γ2α1 onto the AD1b2 axonal compartment. (**E**) Heatmap of synaptic connectivity from MBONs onto each AD1b2 neuron revealing the variability of connectivity across individual AD1b2 neurons. Green dendrogram is a morphological clustering of individual AD1b2 neurons by NBLAST. (**F**) Cartoon summary of dendritic and axoaxonal connectivity from MBONs onto AD1b2 axons. Note that all MBONs also have other currently unknown downstream synaptic partners .

DOI: https://doi.org/10.7554/eLife.43079.014

The following figure supplement is available for figure 8:

**Figure supplement 1.** Comparison of single AD1b2 neurons between Light and Electron Microscopy.

DOI: https://doi.org/10.7554/eLife.43079.015

*Davis, 2009*), activity in which are necessary to induce appetitive and aversive memories respectively (*Aso and Rubin, 2016*; *Claridge-Chang et al., 2009*; *Liu et al., 2012*). These data indicate that LHONs may be upstream of DANs and that innate, hardwired olfactory information may be used to tune MB coding (*Cohn et al., 2015*) or potentially act as a punishment or reward.

To identify downstream sites of convergence between the LH and MB outputs, we performed the above analysis with MBONs, identifying several instances of overlap between MBON and LHON axons (*Figure 8B*, see *Figure 8C'–C'''* for overlaid examples). LHON-to-MBON overlap was sparse, indicating that while regions of convergence and integration exist, many LHONs project to regions that are not directly innervated by the outputs of the MB. Our analysis was concordant with the previous observations using double labelling to identify overlap between PV5a1 (previously named LHON-1), PD2a1/b1, PV5g1/g2 (previously named LHON-2) with MBON axons (*Figure 8B*) (*Aso et al., 2014b*; *Dolan et al., 2018*). As was the case with DANs, the convergence ranged from one to several LHONs interacting with one to many MBONs.

To calculate the total proportion of LHONs which interact with MB-associated neurons, we defined a significant potential interaction as >15% overlap with light microscopy, as validated by previous double labelling experiments (*Dolan et al., 2018*). We found that 70% of LHON axons did not have a significant overlap with any DAN dendrite or MBON axon; only 16% of identified LHONs had a significant overlap with a MBON. Therefore, while we find numerous cases where interactions between LH and MB neurons are likely, we can also rule out such an interaction for the majority of LHONs identified in this study.

A high overlap score between an LHON and MBON could reflect multiple circuit motifs. The neurons may share a common downstream target, form axoaxonic synapses (from LHON to MBON or vice-versa) or a combination of both. Given that AD1b2 converges strongly with several MBONs we wondered if axoaxonic connectivity was a feature of LHON/MBON convergence and if so, what was the direction of information flow. We identified AD1b2 in the EM volume and reconstructed these neurons. We compared the resulting tracing with single-cell light microscopy images to confirm correct identification (*Figure 8-figure supplement 1A'-A''*). We subsequently traced seven AD1b2 neurons (*Figure 8D*, AD1b2#1–7). We first confirmed our light-level pre that AD1b2 neurons receive input from MBON-α1 (*Figure 8D'* and *Figure 7—figure supplement 1D'*). We also identified and traced the three MBONs with the highest overlap with AD1b2 axons in light microscopy, MBON-γ5β'2a (also known as MB-M6, *Figure 8D''*), MBON-α'2 (also known as MB-V4, *Figure 8D'''*) and MBON-γ2α'1 (*Figure 8D''''*). For all three of these MBONs we observed synapses from the MBON onto the axonal segment of the LHON, indicating that the AD1b2 axonal compartment integrates MB output (*Figure 8D–8E*). This input was variable across individual AD1b2 cells (*Figure 8E*). In particular, we identified two highly connected AD1b2 cells (AD1b2#4–5) which had more synapses (85 and 80 synapses respectively) from MBON-γ5β'two than MBON-γ2α'one or MBON-α'2 (*Figure 8E*). Therefore even within an anatomically defined cell type, we observed significant variability in synaptic connectivity, as previously observed for PN synapses onto PD2a1/b1 LHONs (*Dolan et al., 2018*).

These MBONs have other downstream targets and, although the number of synapses is notable, these MBON-to-LHON synapses still make up only a small percentage of the total output synapses of these MBONs (*Figure 8E–F*). Finally, we observed only a small number of synapses from AD1b2 neurons onto MBONs (two for MBON-y2a'1 and 6 for MBON-y5B'2a) indicating that little information flows from AD1b2 LHONs directly to MBONs (data not shown).

MBON-γ2α'1 and MBON-γ5β'2 drive both sleep and valence behaviour in opposite directions, while expressing different neurotransmitters (MBON-γ2α'1 is cholinergic while MBON-γ5β'2 is glutamatergic) (*Aso et al., 2014b*; *Sitaraman et al., 2015*). These data suggest a novel motif for integrating innate and learned sensory representations. AD1b2 LHONs integrate information in three separate domains: From AL PNs on their dendrites within the LH, from MBON-α1 on dendrites outside the LH and from MBON-γ5β'2, MBON-α'two and MBON-γ2α'one on their axon terminals. While it is likely that the dendritic input will be integrated to promote neuronal firing, the axoaxonic input may potentiate (acetylcholine) or gate (glutamate) output onto downstream partners. In summary, a subset of ~30% of LHONs interact with the MB neurons: they can potentially modulate DANs, co-converge onto common downstream targets with MBONs and axoaxonically integrate MB information during olfactory processing.

## Identification of LH neurons that drive valence behaviour

The split-GAL4 toolbox we have developed allows us to specifically manipulate LH cell types in behaving animals. Neurons of the LH are thought to drive innate attraction and repulsion to odours and this has been indirectly demonstrated by MB manipulations (*Heimbeck et al., 2001*; *Min et al., 2013*; *Suh et al., 2004*). To identify LH neurons that might play such roles, we drove an optogenetic effector, ChrimsonR (*Klapoetke et al., 2014*) in individual LH cell types and examined how specific stimulation of these neurons would impact avoidance or approach behaviour (*Figure 9A*). We performed a screen on a subset of LH neurons using mostly 'ideal' split-GAL4 lines and, where possible, multiple lines per cell type (*Figure 9—figure supplement 1A*). We screened 89 split-GAL4 lines (*Figure 9—figure supplement 1A*), covering 50 cell types (and some combinatorial split-GAL4 lines, see *Supplementary file 3*) and identified 15 lines that drove valence behaviour (approach or aversion; *Figure 9B*).

As the empty split-GAL4 line exhibits some attraction to red light (*Figure 9—figure supplement 1B*), we considered aversion to be either an avoidance of illuminated quadrants (*Figure 9C''*) or attenuation of phototaxis (*Figure 9-figure supplement 1C'-C'*). Attraction was classified as a greater attraction to illuminated quadrants *Figure 9C'*. Three lines (each labelling different cell types) induced approach (e.g. *Figure 9C'*), and 12 drove aversion (e.g. *Figure 9C''*) when stimulated. We examined locomotion, turning, upwind and downwind movement in our assay but found no consistent effects across the cell types implicated in driving valence behaviour (data not shown and *Figure 10—figure supplement 1A–C*).

Aversive split-GAL4 lines together labeled six different cell types. Four of these (AV1a1, PV4a1:5, PV5a1, PV5c1) were covered by more than one split-GAL4 line (*Figure 9B*), driving a valence phenotype (*Supplementary file 3*). We focused on the first two cell types for further analysis because all lines containing those cell types showed significant aversion. Although one AV1a1 line (LH728, a 'good' line with off-target expression, see *Figure 9—figure supplement 1F'*) drove strong aversion, most of our avoidance phenotypes only attenuated the attraction to light in controls (*Figure 9B*, *Figure 9—figure supplement 1C*).

To confirm these cell types truly drove repulsion we replicated our hits in different quadrant assay with lower control light attraction, previously used to study MBON valence behaviour (*Aso et al., 2014b*). In this second assay, we observed consistent aversion compared to the control for lines labelling AV1a1 and PV4a1:5 (*Figure 9D*), confirming that these cell types drive avoidance behaviour (see *Figure 9-figure supplement 1D'-D''* for expression patterns of all six aversive split-GAL4 lines).

PV4a1:5 was the most diverse anatomical group isolated by single split-GAL4 drivers, containing in total five anatomically similar cell types, compared to one or two for other split-GAL4 lines targeting a single anatomical group (*Figures 4* and *5*). To narrow down which cell types were driving aversion, we performed extensive single-cell labelling and anatomical clustering on the four PV4a1:5 split-GAL4 lines that were replicated in *Figure 9D*. Unlike other split-GAL4 lines examined, our single-cell data suggested different split-GAL4 lines have overlapping but different combinations of PV4a1:5 cell types (*Supplementary file 4*). However, we interpret these data with caution as the

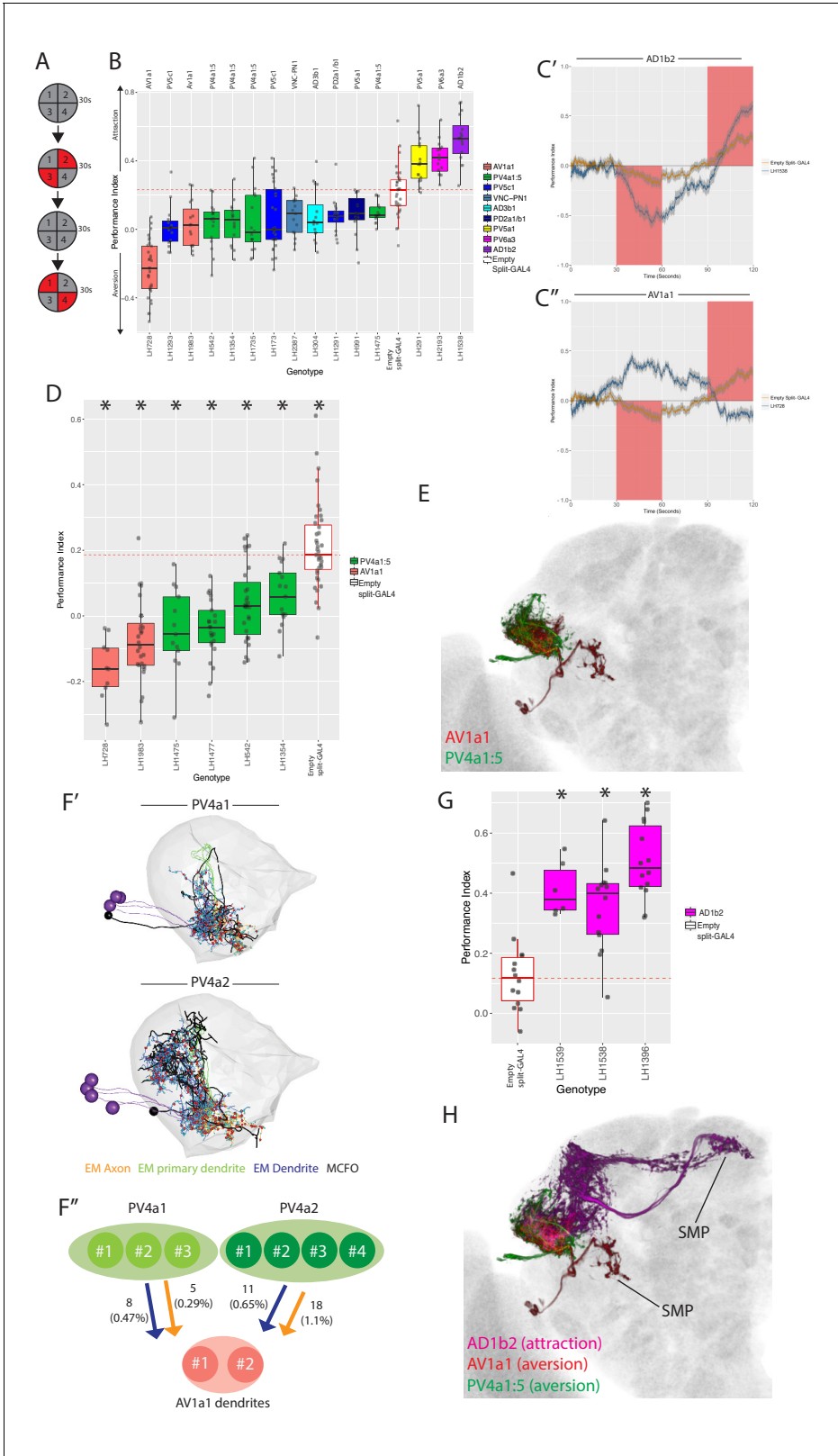

**Figure 9.** Identification of LH neurons that drive aversion and attraction. (**A**) Schematic of the optogenetic stimulation paradigm. Flies are introduced to the chamber and allowed to acclimatize for 30 s followed by illumination of two opposing quadrants for 30 s, a 30 s period of no stimulation, and then a 30 s illumination of the other two opposing quadrants. Distribution of the flies was continuously tracked and used to calculate a preference index (PI). The PI = [(number of flies in Q2 and 3) - (number of flies in Q1 and 4)]/(total number of flies), Q1-4 represents each quadrant. (**B**) Summary box plots

*Figure 9 continued on next page*

*Figure 9 continued*

showing the single value performance index (PI) upon activation of each of the split-GAL4 lines screened that were significantly different from the empty split-GAL4 control (white). The single value PI was calculated by averaging the PIs from the final 5 s of each of the two red-light ON segments of the paradigm (see Materials and methods). Split-GAL4 lines are colour-coded by cell type and the exact cell type is also listed on top of each boxplot. See *Figure 9—figure supplement 1A* for full dataset and statistics. In our assay, Split-GAL4 control flies lacking genomic enhancers to drive effector expression (*Hampel et al., 2015*) had a reproducible attraction to light which was not the case with GAL4 insertion lacking an enhancer (*Figure 9— figure supplement 1B*). Based on this data, we used the Split-GAL4 control as a reference point for calculating optogenetically driven approach or avoidance by LH neurons (red dashed line). (**C'**) A graph of the PI across time, where the split-GAL4 line LH1538 (blue) is plotted against empty split-GAL4 (orange). Activation of LH1538 drives attraction. Red rectangle indicates light ON period. (**C''**) A graph of the PI across time (in seconds), where the split-GAL4 line LH728 (blue) is plotted against empty split-GAL4 (orange). Activation of LH728 drives aversion. Red rectangle indicates light ON period. Line is mean, grey is SEM. (**D**) Optogenetic valence experiments repeated in a different assay (see Materials and methods) for multiple split-GAL4 lines labelling either AV1a1 or PV4a1:5. Asterisk indicates significantly different from the empty split-GAL4 control (p<0.05), n = 10–39 groups. (**E**) Volume rendering showing overlap of AV1a1 (green) and PV4a1:5 (magenta), both are expression patterns from different brains registered to the JFRC2013 template brain. (**F'**) EM reconstruction of PV4 neurons, PV4a1 (top) and PV4a2 (bottom) with presynapses connected to AV1a1 labelled (red spheres). Black neurons represents single PV4 neurons from MCFO data while the EM reconstructed neurons are coloured by axon (orange), primary dendrite (green) and dendritic compartments (magenta), indicating these local neurons have a distinct polarity. The LH is represented by light gray mesh. (**F''**) Summary of EM connectivity from PV4a1/a2 neurons onto AV1a1 dendrites. Numbers represent the number of synapses, numbers in brackets are the percentage of total input from PV4a1 or PV4a2 onto AV1a1 dendrites. (**G**) Optogenetic valence experiments with several split-GAL4 lines labelling cell type AD1b2. Asterisk indicates significantly different from the empty split-GAL4 control (p<0.05), n = 6–14 groups. (**H**) Summary of neurons identified in the valence screen with output regions labelled.

DOI: https://doi.org/10.7554/eLife.43079.016

The following figure supplement is available for figure 9:

**Figure supplement 1.** Further valence screen details and full expression patterns of all split-GAL4 lines discussed in text.
DOI: https://doi.org/10.7554/eLife.43079.017

effector for single neuron labelling was in a different genomic site to the optogenetic effector which can lead to different expression. We found that PV4a4 was present in all split-GAL4 lines while PV4a2 and PV4a3 was present in the majority (*Supplementary file 4*).

As PV4a1:5 was a local LHLN cell type, we wondered how these glutamatergic interneurons could drive aversion. Interestingly we had found potential overlap between PV4a1:5 and AV1a1 in our overlap analysis (*Figure 6D*), indicating that PV4a1:5 could be upstream of this LHON (*Figure 9E*). These data suggest PV4a1:5 to AV1a1 connectivity is excitatory and that AV1a1 may integrate input from LHLNs to produce aversion. To test this inference from our light-level analysis, we identified and reconstructed PV4a cell types in the EM volume. We reconstructed all the cells of the PV4a1 (PVa1#1–3) and PV4a2 (PV4a2#1–3) cell types. In the higher resolution of EM, we observed that these local neurons were polarized with clear dendritic and axonal arbours in the LH (*Figure 9F'*), though both arbors were of mixed polarity. As suggested by our light microscopy-level PV4a1/a2 neurons close interact with AV1a1 dendrites, although this does not necessarily imply synaptic connectivity. To determine this, we examined synapses onto AV1a1 neurons (AV1a1#1–2, which have been identified and reconstructed in *Huoviala et al., 2018*), confirming that PV4a1:5 and AV1a1 are synaptically connected (*Figure 9F''*). Together, PV4a1:2 account for 2.5% of postsynapses onto the AV1a1 dendrite. Although small, we have previously found that MBON-$\alpha$2sc and PD2a1/b1 connectivity is functionally relevant even though MBON-$\alpha$2sc accounted for an average of 2.5% of the post-synapses of these two cell types (*Dolan et al., 2018*). These data suggest PV4a1:5 may drive aversion by stimulating AV1a1.

Finally, we identified three split-GAL4 lines that drive approach when their neurons were optogenetically stimulated. However, only one of these, LH1538 was truly cell type specific (*Figure 9—figure supplement 1E*). This line labeled AD1b2, which we previously identified as integrating input from MBONs (*Figure 8F*). To confirm that stimulation of AD1b2 can induce approach behaviour we tested two additional lines that specifically label this cell type in our valence assay, both of which drove attraction (*Figure 9G*, see *Figure 9—figure supplement 1G* for expression patterns). To summarize, by specifically stimulating subsets of LH neurons we have identified cell types that drive can drive avoidance or approach behaviour when stimulated. However, only a small proportion (~15%) of split-GAL4 lines tested (*Figure 9—figure supplement 1A*) drove such valence behaviour (*Figure 9H*).

## Identification of LH neurons that drive specific motor behaviours

Beyond attraction and aversion, flies display a rich repertoire of innate behaviours in response to olfactory stimulation, such as locomotion modulation, stops and turns (*van Breugel and Dickinson, 2014*; *Budick and Dickinson, 2006*; *Jung et al., 2015*). These responses are likely to depend on hardwired LH circuits, although the role of different LH neurons is unknown.

To increase the range of behaviours we examined, we first examined the behaviour of animals in the optogenetic valence experiments described above. We examined three metrics: forward locomotion, turning and upwind movement during a five second window at the start of the stimulation. Only for forward locomotion did we observe any behaviour statistically different from the empty split-GAL4 control (*Figure 10—figure supplement 1A–C*) and none of these cell types were implicated in valence. For these split-GAL4 lines we repeated the experiments in a flybowl assay with homogenous optogenetic stimulation (*Figure 10A*) (*Simon and Dickinson, 2010*). We successfully replicated a locomotion phenotype with only one cell type, AV6a1, which drives an initial increase in forward locomotion when stimulated (*Figure 10B*). For two split-GAL4 lines labelling AV6a1 (*Figure 10C*, see *Figure 8—figure supplement 1D* for full expression patterns), we found an increase in locomotion relative to the empty split-GAL4 control (*Figure 10D*), confirming that LH neurons can drive changes in motor behaviour unrelated to valence responses.

To more precisely monitor animal motor behaviour, we turned to a single-fly assay in flying animals with a high-resolution recording of wingbeat responses (*Figure 10E*) (*Reiser and Dickinson, 2008*). Tethered animals flew either in the dark or in the presence of a visual stimulus while LH neurons were optogenetically stimulated to determine their effect on behaviour. The visual stimulus was either a stationary stripe, or a closed loop stripe that would respond to the animal's movement (*Figure 10E* and *Figure 10—figure supplement 2*). For each experiment the change in wingbeat amplitude and wingbeat frequency is calculated. As these experiments are low throughput, we chose to focus on four LH cell types. Two of these were bilateral cell types, AD3b1 and PV5k1, which project to the ipsi- and contralateral SLP and central complex respectively (*Figure 2-figure supplement 1B'-B''*). We also examined PD2a1/b1 which are unilateral but have a dual role in attractive behaviour and aversive memory retrieval (*Dolan et al., 2018*) (*Figure 10F–H* and *Figure 10—figure supplement 1E-G* for expression patterns of split-GAL4 lines used).

For all three cell types, we found differential effects of optogenetic stimulation on motor parameters relative to the controls not expressing an optogenetic effector (*Figure 10I–K*). Driving activity in AD3b1 neurons resulted in increased yaw turning by modulation of the left and right wingbeat amplitudes (*Figure 10I*). In contrast, activation of the central complex-projecting PV5k1 resulted in a substantial and persistent decrease in wing beat frequency and therefore flight speed (*Figure 10J*). However, this was persistent only with closed loop visual stimulation (*Figure 10—figure supplement 2*). Although PD2a1/b1 activity is necessary for innate olfactory attraction (*Dolan et al., 2018*), we did not observe a consistent effect of stimulating these neurons in our valence assay (*Figure 9—figure supplement 1A*, LH989). In flight, stimulation of PD2a1/b1 led to persistent increased wingbeat thrust and therefore power, consistent with these neurons driving approach (*Figure 10K*). However, this effect was not persistent in a closed-loop or dark visual paradigm. In summary, the extent of modulation on the flight maneuvers depended on the visual context, suggesting multisensory integration within or downstream of the respective LH neurons (see *Figure 10—figure supplement 2* for full behavioural results across visual conditions). These results imply that individual cell types of the LH can mediate hardwired motor behaviours which in concert might generate diverse responses to a broad palette of odour stimuli. As LH neurons efficiently encode the chemical class of odorants (*Frechter et al., 2019*), this may allow animals to produce similar behaviour to ethologically and chemically similar odours. Moreover, these responses may differ in persistence depending on the sensory context or motor state.

## Discussion

### Summary

Previous work has classified LH cell types with either electrophysiology or calcium imaging (*Fişek and Wilson, 2014*; *Kohl et al., 2013*; *Strutz et al., 2014*). However, due to the lack of sparse driver lines, specific genetic access was not possible for most LH cell types. Two recent studies have

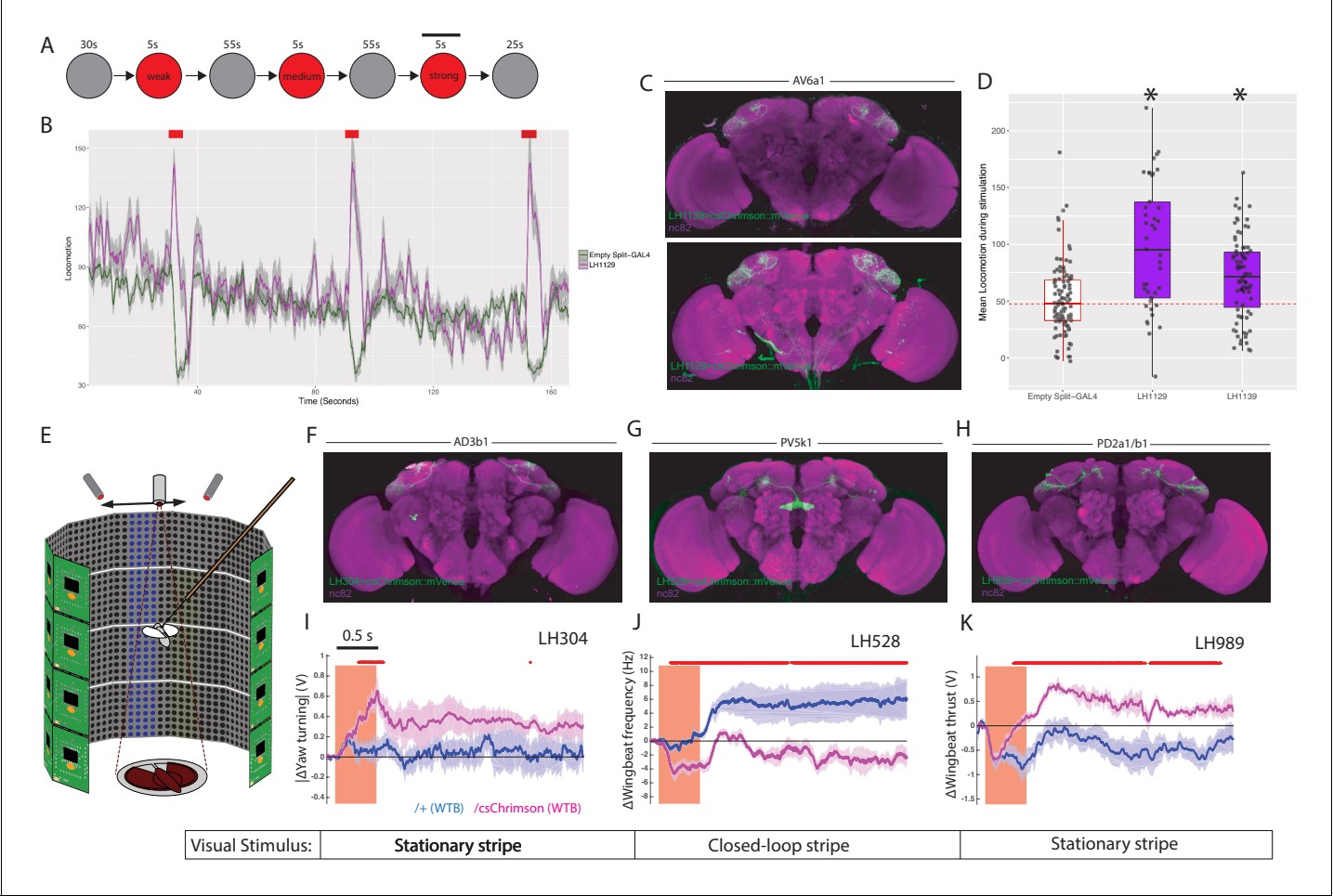

**Figure 10.** LH neurons drive motor behaviours.(**A**) Schematic of the Flybowl optogenetic stimulation paradigm. Flies are introduced to the chamber and allowed to acclimatize for 30 s followed by illumination for 5 s, repeated three times. Each stimulation is of a different intensity, from lowest to highest. The flies were continuously tracked and their movements used to calculate the forward locomotion.The black bar above the 3rd stimulation represents the analysis window for D. (**B**) A graph of the forward locomotion (pixels per second, one pixel is ~0.12 mm) across time, where the split-GAL4 line LH1129 (magenta) is plotted against empty split-GAL4 (green). Line is the mean, grey is the SEM. Activation of LH1129, a split-GAL4 line labelling LH cell type AV6a1, drives an increase in forward locomotion compared to control. Red rectangle indicates light ON period of increasing intensity. (**C**) Z-projection of the brain expression patterns of the two split-GAL4 lines for cell type 7A, LH1139 (top) and LH1129 (bottom). Note minor trachea labelling in LH1129. All split-GAL4 lines are driving csChrimson::mVenus (green), the neuropil stain is nc82 (magenta). See ***Figure 10—figure supplement 1*** for full brain and nerve cord expression patterns. (**D**) Boxplots analysing the mean locomotion during the strongest stimulation window of five seconds (locomotion measured in pixels per second, one pixel is ~0.12 mm). Each dot represents the behaviour of a single fly. Asterisk indicates significantly different from the empty split-GAL4 control. (**E**) Schematic of the flying optogenetic stimulation paradigm. Animals are tethered in a flight area. Flying animals have either no visual stimulus, a stationary bar or a closed loop blue bar. During flight optogenetic stimulation is driven by LED stimulation bilaterally. Schematic was modified with permission. (**F–H**) Z-projection of the brain expression patterns of the three split-GAL4 lines used in the flight assay: (**F**) LH304, (**G**) LH528 and (**H**) LH989. All split-GAL4 lines are driving csChrimson::mVenus (green). Note for all expression pattern images, the neuropil stain is nc82 (magenta). See ***Figure 10—figure supplement 1*** for full brain and nerve cord expression patterns. (**I**) The absolute value of yaw (turning) over time (seconds) of single LH304 split-GAL4 flies crossed to experimental (magenta) and control genotypes (dark purple). This represents turning. (**J**) The wingbeat frequency over time (seconds) of single LH528 split-GAL4 flies crossed to experimental (magenta) and control genotypes (dark purple). The reduction in wingbeat frequency represents a reduction in speed and persists after stimulation. (**K**) The wingbeat thrust over time (seconds) of single LH989 split-GAL4 flies crossed to experimental (magenta) and control genotypes (dark purple). The increase in thrust represents increased power. WTB = Wild Type Berlin background strain. In I-K, lines are mean traces from all animals, coloured shades around the lines are standard errors of the mean across the animals (n = 6 for each genotype). The rectangle pink shades indicate the periods of activation light on. The red +marks above each plot indicate the time points at which the CsChrimson traces are significantly different from the negative control traces (Wilcoxon rank sum test, p<0.05). Time scale for 0.5 s is on the top of the first stimulation period.

DOI: https://doi.org/10.7554/eLife.43079.018

The following figure supplements are available for figure 10:

*Figure 10 continued*

**Figure supplement 1.** Expression patterns of split-GAL4 lines used for single fly behaviour.

DOI: https://doi.org/10.7554/eLife.43079.019

**Figure supplement 2.** Full behavioural data for flying optogenetic experiments.

DOI: https://doi.org/10.7554/eLife.43079.020

identified more driver lines labelling LH neurons but these are broad and suitable mostly for electro-physiology and dye-filling identification of individual cells (*Frechter et al., 2019*; *Jeanne et al., 2018*). In this study, we generated split-GAL4 lines that enable anatomical analyses and specific neu-rogenetic control of LH cell types. We used these reagents to generate an atlas of identified LH neu-rons, defined their major neurotransmitter and polarity; classifying most cell types into LHONs, LHLNs and LHINs. Connectomic analysis of neural circuits in EM volumes is providing an unprece-dented window into the structure of neural circuits, especially in *Drosophila* (*Dolan et al., 2018*; *Eichler et al., 2017*; *Ohyama et al., 2015*; *Schlegel et al., 2017*; *Takemura et al., 2013*, *Takemura et al., 2017*). To facilitate connecting our genetic tools with EM data we provided traced backbones of neurons from many identified LH cell types in a whole-brain EM volume (*Zheng et al., 2018*), in addition to light-level single-neuron labelling (*Nern et al., 2015*). These resources will allow future studies to correlate functional and connectomic data to determine how the LH gener-ates olfactory behaviour.

We used our driver lines to identify major output zones and convergence sites of LH neurons. The diversity of inter- and intra-regional connections demonstrates the complexity of the LH and the innate olfactory circuitry, even in the relatively 'simple' brain of *Drosophila*. In terms of output, our anatomical analysis identified the SLP as the next major site of olfactory processing, although LHONs projected to many other brain regions, with a broader range than MBONs (*Aso et al., 2014a*). Across identified cell types, we did not identify any LHONs which project to the ventral nerve cord, suggesting at least one more layer of processing before motor output. Within the LH, previous stud-ies characterized one population of GABAergic local neurons (*Fişek and Wilson, 2014*). Other than these and MB associated neurons however, the neurotransmitter profiles of neurons in the proto-cerebrum, were unknown. We identified populations of LH neurons which were cholinergic, GABAer-gic and/or glutamatergic. We could cluster LHONs, LHINs and LHLNs into groups based on the sign of their neurotransmitter. We found several distinct populations of both GABAergic and glutamater-gic cells which were local to the LH, some potentially interacting with many LHON dendrites. This indicates that both lateral excitation and inhibition may exist in the LH. We also found that the LH integrates visual, gustatory (*Kim et al., 2017*), thermosensory (*Frank et al., 2015*) and mechanosen-sory (*Patella and Wilson, 2018*) input in a restricted, ventral region.

The LH and MB are thought to mediate innate and learned behaviour respectively, yet their inter-actions remain little understood. Several MBONs project to the LH and our work here has identified potential downstream targets. Indeed, one of our predictions (connectivity from MBON-α2sc to PD2a1/b1 neurons) has been validated and shown to be necessary for innate and learned behaviour (*Dolan et al., 2018*). In addition we found that AD2b2 neurons extend their dendrite outside the LH and receive MB input. Previous studies have identified three instances of MB and LHON axonal con-vergence (*Aso et al., 2014b*; *Dolan et al., 2018*), however it was unclear if this was the case for all LHONs. We systematically compared overlap for all the LHONs identified in our split-GAL4 screen. We find ~30% of LHONs converge with MB-associated neurons, several of which appear placed to interact with more than one DAN and/or MBON. Therefore while this is a significant circuit motif, it does not explain all LHON projections. We find AD1b2 interact extensively with three MBONs in a unique axoaxonic integration motif, and stimulation of AD1b2 neurons drives approach behaviour when stimulated. Although the MB and LH both receive input from the AL, the coding logic of these two regions is strikingly different (*Fişek and Wilson, 2014*; *Gruntman and Turner, 2013*; *Honegger et al., 2011*) and the new tools we describe here will greatly facilitate studies of how these regions together orchestrate behaviour.

To apply these reagents experimentally, we used cell type specific optogenetic activation and identified several LH neurons whose activation drives behaviour. We identified two cell types (one LHON and one LHLN) which drive aversion, one LHON which drives attraction and several LHONs whose stimulation leads to changes in motor behaviours. Our optogenetic screen identified only 3/

50 LH cell types that could alone consistently drive changes in valence. This was a lower proportion than in the MB, where 6/20 MBON cell types tested were implicated solely in attraction and aversion rather than specific motor parameters (*Aso et al., 2014b*; *Owald et al., 2015*).

## Coverage of split-GAL4 lines across Lateral Horn cell types

In total, our split-GAL4 lines covered 82 different cell types in the LH, although 24 of these were consistently colabeled in split-GAL4 lines. In addition, five anatomical cell types (PV4a1:5) occur in overlapping but disjoint split-GAL4 lines. Anatomical analysis of single neurons labeled with either stochastic genetic labelling (*Chiang et al., 2011*) or dye fills during recording (*Frechter et al., 2019*) identified ~165 different 'core' cell types in the LH (*Frechter et al., 2019*), that is cell types that seemed to have a majority of their dendrite within the LH and in the vicinity of olfactory uniglomerular PNs. Core LH cell types excluded LHINs and any neurons with a low fraction of their dendritic arbor overlapping the PN terminals bounded by the LH (*Frechter et al., 2019*). To calculate our coverage compared to these datasets, we used the same definition to categorize LH cell types identified in split-GAL4 lines as core or non-core. For LHON and LHLN cell types (68 in total), 63 were classified as core LH cell types (see *Supplementary file 1*). Based on this definition, our split-GAL4 collection covers 63/165 or ~38% of known core LH cell types.

For the five cell types not defined as core, connectomic analysis indicates that they receive substantial input from multiglomerular PNs and/or some level of input from canonical uniglomerular PNs (A.S.B. and G.S.X.E.J., in preparation).

Although this 38% is lower than estimates of coverage for the MB, which exceeds 80% (*Aso et al., 2014a*; *Takemura et al., 2017*), it is similar to the estimated coverage achieved to date (30–50%) for split-GAL4 lines targeting descending interneurons from the brain to VNC (*Namiki et al., 2018*). There are several possible hypotheses for these differences. The first is simply that there appear to be fewer MB cell types, for example 23 MBONs (*Aso et al., 2014a*; *Takemura et al., 2017*), than LH or descending interneurons. The second is that the dense, compartmental arborization of most MB neurons makes them easier to identify in lines containing many cell types, which are the starting point for split-GAL4 screens. Finally, it may well be that neurons in different brain regions have more or less distinct transcriptomic and epigenetic profiles and that this impacts the genetic isolation of their constituent cells. We would also emphasize that full and comprehensive reconstruction of all these neuronal classes from whole brain EM data will likely reveal additional cell types not discovered in our annotations of genetic data (*Takemura et al., 2017*; *Zheng et al., 2018*).

## Advantages and limitations

This split-GAL4 driver line resource will now allow investigators to manipulate many LH neurons with cell type specificity. Driver lines can be used for imaging, functional activation or silencing experiments with genetically-encoded optogenetic or thermogenetic tools (*Venken et al., 2011*). However, we caution that the final expression pattern of any driver line also depends on the insertion site of the effector transgene (*Aso et al., 2014a*; *Pfeiffer et al., 2010*); expression patterns should be verified for effectors at locations other than attP18 (the site we used during our screening).

In the course of this work we have produced large-scale anatomical maps of overlap between different populations of LH and MB neurons. This is a rapid approach to generate hypotheses for synaptic connectivity and circuit motifs. However, all potential synaptic interactions are subject to both registration error and biological variability across different brains (*Jefferis et al., 2007*). While we validated several highly overlapping pairs with EM reconstruction, even neurons that are very close in space may not be synaptically connected and this must be confirmed with connectomics or physiology.

During our behavioural studies, we focused only on phenotypes that were consistent across multiple 'ideal' split-GAL4 lines for the same cell type. Indeed one of the four PD2a1/b1 lines tested had an aversive phenotype when activated, contrary to our previous data (*Dolan et al., 2018*). This is most likely due to off-target expression in the VNC, as described (*Dolan et al., 2018*) and we note that stimulation of PD2a1/b1 neurons, using a different split-GAL4 line, in flying animals drove attraction (*Figure 10*). Despite this, not all differences between split-GAL4 lines ostensibly targeting the same cell type should be discarded. Some of the differences in phenotype between split-GAL4

lines targeting the same cell type could be due to differences in expression strength or simply experimental variation during a large-scale screen (that was not designed to identify small effect sizes). Even within an anatomically and genetically defined cell type, neurons can have different connectivities (*Dolan et al., 2018*). Therefore it is quite possible that we have discounted viable phenotypes in our analysis and subsequent replication experiments. Finally, it also possible that the relatively low number of phenotypes in our optogenetic activation screen is because activation of just one of >150 LH cell types is often too selective to produce strong behavioural consequences, especially in the absence of an olfactory stimulus. Future studies will be able to combine our split-GAL4 lines to control multiple LH cell types (*Aso et al., 2014b*) or use transsynaptic labelling to control large populations postsynaptic to specific cell types (*Talay et al., 2017*).

## Functions of the Lateral Horn

In this paper, we directly demonstrate the existence of a diversity of genetically defined cell types with highly stereotyped dendritic arbours in the LH. Additionally, we demonstrate that stimulation of LH neurons can drive valence or motor behaviours. While our gain-of-function experiments do not alone demonstrate the role of the LH in innate behaviour, when combined with published data there is now clear anatomical, functional and behavioural evidence that the LH mediates instinctive olfactory responses. Firstly, anatomical and functional experiments demonstrate stereotyped connectivity between PNs and LH cell types between animals (*Fişek and Wilson, 2014*; *Frechter et al., 2019*; *Huoviala et al., 2018*; *Jeanne et al., 2018*; *Jefferis et al., 2007*), implying a role in the generation of innate behaviour in contrast to the nearly random connectivity between PNs and the MB (*Caron et al., 2013*; *Gruntman and Turner, 2013*). Neuronal silencing experiments demonstrate that abolition of KC neurotransmission leads to a silencing of memory and a reset to innate olfactory responses (*Heimbeck et al., 2001*; *Parnas et al., 2013*). In addition to this work, two contemporaneous studies from our group have used a subset of the new split-GAL4 lines to interrogate the role of this region in specific learned and innate olfactory behaviours (*Dolan et al., 2018*; *Huoviala et al., 2018*). While *Dolan et al. (2018)* show that PD2a1/b1 LHONs are required for both innate attraction and aversive memory recall, the memory recall phenotype seems best understood as the modulation by a mushroom body output pathway of a hardwired pathway required for naive behaviour

Given the extensive interactions between LH and MB identified in this study we propose these 'horizontal' pathways (e.g. MBONs projecting to LH, LHON to DAN) orchestrate additional functions such as memory retrieval, provide categorical information and MB modulation. We suggest that LH-to-MB information flow (e.g via LHON-to-DAN synapses) may also explain why some MBONs exhibit valence-specific responses (*Hige et al., 2015b*), a hypothesis which can now be tested using the split-GAL4 lines developed here.

The tools described in this study provide a critical resource for cell type specific dissection of the LH. We found that stimulation of AV1a1 and PV4a1:5 drive avoidance while silencing AV1a1 abolishes the response to geosmin (an ecologically relevant aversive odour) in an oviposition assay (*Huoviala et al., 2018*). This suggests AV1a1 LHONs may be a major pathway for ethologically-relevant aversion in the fly brain and that PV4a1:5 interneurons may pool other inputs to drive aversion via this pathway.

We also found that AD1b2 LHONs drive approach behaviour when activated. These cholinergic neurons project axons to the SMP and have a distinctive additional dendritic projection extending out to the SMP/SIP. AD1b2 integrates MBON input in both an axodendritic and axoaxonic manner, including two MBONs which bidirectionally drive valence behaviour. Interestingly, we previously identified another LHON (PD2a1/b1) involved in innate attraction (*Dolan et al., 2018*). PD2a1/b1 also received MBON input onto both its dendrite and axon, and both AD1b2 and PD2a1/b1 receive axonal input from MBON-α′2 a memory-relevant MBON. These parallels suggest both a shared circuit motif and a relation between (innate) approach behaviour and memory.

In general however, naive olfactory responses are more diverse than attraction or aversion, and olfactory stimulation can drive changes in locomotion/flight speed, stopping and turning or exploratory behaviour (*van Breugel and Dickinson, 2014*; *Budick and Dickinson, 2006*; *Jung et al., 2015*; *Keller and Vosshall, 2007*). We identified one LH cell type which drives forward locomotion when stimulated, but does not impact valence behaviour. Much olfactory sensation occurs during flight and by recording the wingbeat responses of flying *Drosophila* before and after optogenetic stimulation we could identify LH neurons that modulate different parameters (turning, wing thrust and

wingbeat frequency) in flying animals. Interestingly, the effects of stimulating these neurons had different impacts on behaviour depending on visual stimulus (see multimodal integration below) while the persistence of these effects post-stimulation also varied. Therefore, different LH neurons likely drive different motor programs at various timescales. This diversity in downstream functions may explain why the LH has a large number of cell types which show distinct but frequently overlapping odour responses or pool different PN channels (*Frechter et al., 2019*; *Jeanne et al., 2018*). Behavioural responses to odours (e.g. exploratory behaviour) may be a composite of different motor programs (e.g. locomotion increase, decrease turning) where each program is driven by a small assembly of different LH cell types. However, we found only one LH neuron projecting to the sensorimotor integration circuitry of the central complex. Taken together our results imply that LH output is not immediately sent to motor neurons, likely it is integrated downstream with other information such as memory or internal state. This model is supported by the observation of specific impacts on motor behaviour by manipulations of different olfactory sensory neurons, although such manipulations also impact downstream MB circuits (*Jung et al., 2015*).

We found extensive overlap between a minority of LHONs and MB neurons, and an additional function of the LH may be to modulate and monitor the distributed coding in the MB. LHONs may modulate or even implant memories via DANs (*Aso and Rubin, 2016*) and the axoaxonic integration of MBON inputs by LHONs may occur more generally (*Dolan et al., 2018*). Higher olfactory neurons downstream of MB and LH likely read out olfactory information from these two different coding regimes to integrate decorrelated and hardwired olfactory representations.

In addition to olfactory responses, our data also demonstrates the LH integrates multimodal input. This non-olfactory sensory information could provide context for the innate olfactory processing system which may be relevant for naive responses to odours or for pheromonally-driven behaviours such as courtship or aggression (*Kurtovic et al., 2007*; *Wang and Anderson, 2010*). Multimodal sensory input may also provide a route for direct integration of context and olfactory stimulation. Indeed, we found the effects of optogenetic stimulation of LH neurons differed depending on the visual stimulation displayed to flying animals.

In future experiments, the split-GAL4 tools described here will help investigators to determine the exact functions of the LH during olfaction at single cell type resolution. While the functions of individual LH neurons can range from aversion to memory retrieval, many LH neurons do not appear to be solely involved in avoidance or attraction. Higher-resolution behavioural assays will be needed (*Anderson and Perona, 2014*). These will identify the subtle olfactory responses beyond avoidance or attraction (*Jung et al., 2015*) and can be combined with connectomic information to design precise loss-of-function experiments.

## Conclusions

Both the mammalian and insect olfactory systems split into two parallel processing tracks, one hardwired or chemotropic and one distributed or random with functional roles in innate and learned behaviour respectively (*Choi et al., 2011*; *Li and Liberles, 2015*; *Miyamichi et al., 2011*; *Root et al., 2014b*; *Sosulski et al., 2011b*; *Su et al., 2009*). This striking similarity suggests we may be able to obtain important and general insights into how olfactory perception is transformed into action using the simpler nervous system of *Drosophila*. Using the split-GAL4 lines generated here we were able to provide a map of the inputs and outputs of this region, identifying many previously unknown features, such as multimodal input and extensive interactions with the MB. In addition we have identified LH neurons that are sufficient to drive valence behaviour or specific motor movements, providing the first functional evidence that this hardwired region can direct diverse olfactory responses. This work represents a step towards a full model of how olfactory stimuli are processed in the *Drosophila* brain, from sensory input to motor output.

## Note added in proof

*Varela et al., 2019* have recently characterised two LH cell types with a role in behavioural responses to $CO_2$.

# Materials and methods

## Key resources table

| Reagent type (species) or resource | Designation | Source or reference | Identifiers | Additional information |
|---|---|---|---|---|
| Genetic reagent (D. melanogaster) | Drosophila split-GAL4 Line, LH989 | *Dolan et al., 2018* | LH989 | http://splitgal4.janelia.org/cgi-bin/splitgal4.cgi |
| Genetic reagent (D. melanogaster) | Drosophila split-GAL4 Line, LH991 | *Dolan et al., 2018* | LH991 | http://splitgal4.janelia.org/cgi-bin/splitgal4.cgi |
| Genetic reagent (D. melanogaster) | Drosophila split-GAL4 Line, LH2447/MB077B | *Aso et al., 2014a* | LH2447/MB077B | http://splitgal4.janelia.org/cgi-bin/splitgal4.cgi |
| Genetic reagent (D. melanogaster) | Drosophila split-GAL4 Line, LH2449/MB380B | *Aso et al., 2014a* | LH2449/MB380B | http://splitgal4.janelia.org/cgi-bin/splitgal4.cgi |
| Genetic reagent (D. melanogaster) | Drosophila LH split-GAL4 Line, LH split-GAL4 lines generated in this study | this study, see *Supplementary file 1* | LHXXXX | http://splitgal4.janelia.org/cgi-bin/splitgal4.cgi |
| Genetic reagent (D. melanogaster) | csChrimson Reporter/Optogenetic effector, 20xUAS-csChrimson::mVenus (attP18) | *Klapoetke et al., 2014* | 55134, Bloomington Stock Center | |
| Genetic reagent (D. melanogaster) | Polarity reporter, w; +; 3xUAS-Syt::smGFP-HA (su(Hw)attP1), 5xUAS-IVS-myr::smGFP-FLAG (VK5) | *Aso et al., 2014a* | | |
| Antibody | Mouse anti-Brp, monoclonal | DSHB, University of Iowa, USA | nc82 | 1/30 |
| Antibody | Rabbit anti-GFP, polyclonal | Life Technologies | A-11122 | 1/1000 |
| Antibody | Goat anti-Rabbit AF488 | Life Technologies | A-11008 | 1/400 |
| Antibody | Goat anti-Rabbit AF568 | Life Technologies | A-11004 | 1/800 |
| Antibody | Rat anti-FLAG, monoclonal | Novus Biologicals | NBP1-06712SS | 1/200 |
| Antibody | Rabbit anti-HA, monoclonal | Cell Signalling Technologies | C29F4 | 1/300 |
| Antibody | ATTO 647N Goat α-Rat IgG | Rockland | 612-156-120 | 1/300 |
| Antibody | Cy3 Goat α-Rabbit | Jackson Immuno Research | 111-165-144 | 1/1000 |
| Antibody | Cy2 Goat α-Mouse | Jackson Immuno Research | 115-225-146 | 1/600 |
| Antibody | Mouse α-V5, polyclonal | AbD Serotec | DL550 | 1/500 |
| Antibody | AF488 Goat α-Mouse | Life Technologies | A-11001 | 1/400 |
| Antibody | AF594 Donkey α-Rabbit | Jackson Immuno Research | A-21207 | 1/500 |

*Continued on next page*

*Continued*

| Reagent type (species) or resource | Designation | Source or reference | Identifiers | Additional information |
|---|---|---|---|---|
| Chemical compound, drug | Paraformaldehyde | Electron Microscopy Sciences | #15713 | |
| Software, algorithm | Fluorender | *Wan, 2012* | | http://www.sci.utah.edu/software/fluorender.html |
| Software, algorithm | Fiji | *Schindelin et al., 2012* | | http://fiji.sc/ |
| Software, algorithm | CMTK | | | https://www.nitrc.org/projects/cmtk/ |
| Software, algorithm | CATMAID | *Schneider-Mizell et al., 2016* | | www.catmaid.org |
| Software, algorithm | R | | | https://www.r-project.org/ |
| Software, algorithm | nat | | 10.5281/zenodo.1136106 | http://jefferis.github.io/nat/ |
| Software, algorithm | elmr | *Zheng et al., 2018* | 10.5281/zenodo.1401050 | https://jefferis.github.io/elmr/ |
| Software, algorithm | lhns | This study and *Frechter et al., 2019* | 10.5281/zenodo.2631765 | https://github.com/jefferislab/lhns |
| Software, algorithm | 2018lhsplitcode | This study | | https://github.com/jefferislab/2018lhsplitcode |
| Software, algorithm | flywatch | This study | 10.5281/zenodo.2631286 | https://github.com/jefferislab/flywatch |
| Software, algorithm | LHLibrary | *Frechter et al., 2019* | 10.5281/zenodo.2635574 | https://github.com/jefferislab/LHlibrary |
| Software, algorithm | NMF | *Gaujoux and Seoighe, 2010* | | https://cran.r-project.org/package=NMF |

## Molecular biology

Hemidriver lines were created using gateway cloning as previously described (*Dionne et al., 2018*). All transgenic fly lines were generated by either Bestgene Inc or Genetic Services, Inc. Note that two Split-GAL4 lines for cell type PD2a1/b1 (LH989 and LH991) have been previously published (*Dolan et al., 2018*). Three lines were developed as part of a split-GAL4 screen of the MB (*Aso et al., 2014a*). Two split-GAL4 lines were previously published and are renamed here: LH2447 (MB077B) and LH2449 (MB380B) (*Aso et al., 2014a*)

## Drosophila husbandry

All experiments in the paper used female flies. Standard techniques were used for fly stock maintenance and construction. For imaging and immunohistochemistry flies were raised at 25˚C on standard *Drosophila* food. For optogenetic experiments flies were grown on standard *Drosophila* food supplemented with retinal (see below).

## All published *Drosophila* effector stains used in this study are listed below

- *20xUAS-csChrimson::mVenus (attP18)* For split-GAL4 screening and neuronal visualization. Bloomington Stock Number: 55134 (*Klapoetke et al., 2014*).
- *w; +; 3xUAS-Syt::smGFP-HA (su(Hw)attP1), 5xUAS-IVS-myr::smGFP-FLAG (VK5)* Polarity and membrane marker (*Aso et al., 2014a*).

- *hsFlp2::PEST (attP3);+; 10XUAS-FRT > STOP > FRT-myr::smGFP-HA (VK00005), 10XUAS-FRT > STOP > FRT-myr::smGFP-V5-THS-10XUAS-FRT > STOP > FRT-myr::smGFP-FLAG (su(Hw) attP1)/TM3, Sb* Reporter for MultiColor FlpOut (MCFO) (*Nern et al., 2015*).
- *w;; R22C06-GAL4 (attP2)* Driver line for labelling mALT temperature PNs analyzed in *Figure 7* (*Frank et al., 2015*; *Jenett et al., 2012*).
- *w;; R84E08-GAL4 (attP2)* Driver line for labelling t3ALT temperature PNs analyzed in *Figure 7* (*Frank et al., 2015*; *Jenett et al., 2012*).
- w;; R60H12-GAL4 (attP2) Driver line for labelling t5ALT temperature PNs analyzed in *Figure 7* (*Frank et al., 2015*; *Jenett et al., 2012*).
- w;; R95C02-GAL4 (attP2) Driver line for labelling lALT temperature PNs analyzed in *Figure 7* (*Frank et al., 2015*; *Jenett et al., 2012*).
- w;; R11H09-GAL4 (attp2) Driver line for labelling the TPN2 gustatory PNs analyzed in *Figure 7* (*Kim et al., 2017*; *Jenett et al., 2012*).
- w;; VT57358-GAL4 (attp2) Driver line for labelling the TPN3 neurons analyzed in *Figure 7* (*Kim et al., 2017*; *Kvon et al., 2014*).
- *20xUAS-ChrimsonR::mVenus (attP18)* For optogenetic stimulation experiments. Bloomington Stock Number: 55134 (*Klapoetke et al., 2014*).
- *20xUAS-ChrimsonR::mVenus (attP18)* in Wild-Type Berlin genetic background, for increased flight behaviour. For crossing to Split-GAL4 lines for single-fly optogenetic experiments
- Wild-Type Berlin. Used as a non-effector control for crossing to Split-GAL4 lines for single-fly optogenetics experiments (*Colomb and Brembs, 2014*).

## The Split-GAL4 lines generated and used by this study were of the form

- *w*; Enhancer1-*ADp65 (attP40);* Enhancer2-*ZpGAL4DBD (attP2)* as described previously (*Pfeiffer et al., 2010*; *Luan et al., 2006*; *Aso et al., 2014a*). One exception was LH2223 which has the AD and DBD in the same site (attP2). See *Supplementary file 1* for full list of LH Split-GAL4 lines.

## Split-GAL4 hemidriver combination screening

To generate these reagents, we first identified enhancers (from two enhancer GAL4 collections, see below) that drive expression in each of the LH cell types. In a previous study we developed a nomenclature system for the neurons of the LH, a hierarchical classification based on primary neurite tract, then axonal and dendritic projections (*Frechter et al., 2019*). This analysis, which was based on annotation of broad driver lines and single cell data from stochastic genetic labelling, identified a total of 31 primary neurite tracts and 165 anatomical cell types. We used this nomenclature as a starting point to search publicly available enhancer fusion GAL4 collections and identify enhancer expression patterns that label the same LH cell type. We screened LH neurons in 3,378 GAL4 lines from the GMR collection (*Jenett et al., 2012*) and 2,916 GAL4 lines from the Vienna Tile collection (*Kvon et al., 2014*). We identified 63 of these known LH anatomical cell types and several new LHINs. Although these driver lines more successfully labelled LH neurons than previous studies, very few of these GAL4 lines were sparse enough for functional analysis. We next used these annotations to generate 99 hemidriver lines, where the GAL4 was replaced with an AD or DBD fused to a particular enhancer and merged this collection with hemidriver lines generated by previous and ongoing studies (*Dionne et al., 2018*; *Aso et al., 2014b*; *Bidaye et al., 2014*; *Tuthill et al., 2013*; *Wu et al., 2016*). By combining cognate hemidrivers (ie. AD x DBD) targeting the same groups of LH neurons in the same animal, we could specifically label small groups of neurons by Split-GAL4 intersection.

The Split-GAL4 hemidriver lines selected were based on the GAL4 expression pattern using the same enhancer. The collection of hemidriver lines are a resource developed at Janelia Research Campus. Each Split-GAL4 line consists of two hemidrivers, the p65ADZp in attP40 and the ZpGAL4DBD in attP2. Potential Split-GAL4 lines were selected by choosing two hemidriver lines (an AD and DBD) whose GAL4 patterns putatively contained the same cell types. The lines were screened by combining these two hemidrivers in the following manner: 20xUAS-csChrimson::mVenus (attP18)/w; Enhancer-p65ADZp (attP40)/+; Enchancer-ZpGAL4DBD (attP2)/+

The brains of Split-GAL4 driver combinations were dissected and mounted for manual screening of endogenous fluorescence of mVenus. This preliminary step without immunohistochemical amplification allowed for the selection of strongly expressing lines. Split-GAL4 combinations were screened

using epifluorescence mode on a LSM710 confocal microscope (Zeiss) using a Plan-Apochromat 20x/ 0.8 M27 objective. Combinations were screened for lines that labelled lateral horn (LH) cell types both strongly and specifically. All lines deemed sufficiently sparse by endogenous mVenus signal underwent whole-mount immunohistochemistry and confocal imaging (see Protocol one below). Split-GAL4 combinations with favourable expression patterns were double balanced and combined in the same fly to make a stable stock which was used for further characterization (Dionne et al. 2017). Each intersection was screened once (n = 1 brain), double balanced if selected and its expression pattern was manually compared to further polarity, neurotransmitter, MCFO and csChrimson stainings (total of at least n = 11).

## Split-GAL4 line characterization (Polarity, Single neurons, Neurotransmitter)

All polarity images are representative of at least n = 3 independent images. For analysis of neuronal polarity, stabilized Split-GAL4 lines were crossed to the polarity effector line (see above) for differentially labelling presynapses and neuronal membrane (*Aso et al., 2014a*). These flies were dissected and underwent whole-mount immunohistochemistry and confocal imaging (see Protocol two below). Due to position effects, a small number of split-GAL4 lines were not specific when crossed to the polarity effector line (as this is in a different site to the csChrimson::mVenus used during screening). In these cases, we crossed the split-GAL4 line again to the Chrimson effector. We performed immunohistochemistry for csChrimson::mVenus and nc82 but used our 63x imaging protocol (see below) for high-resolution imaging.

Our categorization of LHON and LHIN is based two criteria:

1. A qualitative examination of the Syt::HA stain in the LH versus other regions. Although we did see incomplete polarisation (see *Figure 1—figure supplement 1* for example), it was typically weighted heavily in one direction.
2. A qualitative examination of synapses in our traced single neurons reconstructed in the EM dataset

One cell type, PV2d1, appears to have the majority of its presynaptic signal in the LH (*Figure 6—figure supplement 1B*). However, in this case a qualitative examination of presynapses in EM strongly implied these are actually LHONs. This discrepancy may be due to overexpression of Syt:: HA or ectopic synapses due to over expression of Syt itself.

For single cell labelling of LH neurons we used the MultiColor FlpOut (MCFO) technique (*Nern et al., 2015*). For MultiColor FlpOut (MCFO) experiments, the MCFO stock (see above) was crossed to a Split-GAL4 line. Flies were collected after eclosion, transferred to a new food vial and incubated in a 37°C water bath for 20–25 min. These flies were dissected and underwent whole-mount immunohistochemistry and confocal imaging (see Protocol three below).

All neurotransmitter stainings are representative of at least n = 2 stainings. For neurotransmitter staining, we performed the stainings using as previously described (*Dolan et al., 2018*). Split-GAL4 lines were crossed to *20xUAS-csChrimson::mVenus (attP18)* and underwent whole-mount immunohistochemistry and confocal imaging (see Protocol four for staining for glutamatergic and cholinergic neurons and Protocol five for staining for GABA reactivity.)

## Immunohistochemistry and imaging

In this study we used several different protocols for whole mount immunohistochemistry and imaging as indicated in the appropriate methods text. All imaging was performed on a LSM710 confocal microscope (Zeiss). Full step-by-step protocols can be found at https://www.janelia.org/project-team/flylight/protocols. For neurotransmitter staining experiments, bouton overlap was determined manually for VGlut or ChAT staining while a GABA-positive stain was determined by examining the cell body overlap.

## Protocol 1: Initial Split-GAl4 characterization

The following primary antibodies were used:

- 1/30 Mouse α-Brp (nc82) (DSHB, University of Iowa, USA)
- 1/1000 Rabbit polyclonal α-GFP (A-11122) (Life Technologies, now Thermofisher Waltham, MA, USA)

The following secondaries were used:

- 1/400 AF488 Goat α-Rabbit (A-11008) (Life Technologies, now Thermofisher Waltham, MA, USA)
- 1/800 AF568 Goat α-Mouse (A-11004) (Life Technologies, now Thermofisher Waltham, MA, USA)

Following the standard immunohistochemistry protocol the brains were fixed again in 4% Paraformaldehyde (Electron Microscopy Sciences, Hatfield, PA) for four hours at room temperature. The brains were mounted on poly-L-lysine-coated cover slips and dehydrated through a series of ethanol baths (30%, 50%, 75%, 95%, and 3 × 100%) each for 10 min. Following dehydration they were submerged in 100% Xylene three times for 5 min each. Samples were embedded in DPX (DPX; Electron Microscopy Sciences, Hatfield, PA). Whole mount brain and VNCs were imaged using a Plan-Apochromat 20x/0.8 M27 objective (voxel size = 0.56 × 0.56×1.0 μm; 1024 × 1024 pixels per image plane). All images of brains expressing csChrimson::mVenus were visualised in 20x mode. Note that this protocol was also used to get extra, confirmatory images of split-GAL4 lines used in behaviour.

## Protocol 2: Polarity

The following primaries were used:

- 1:30 Mouse α-Brp (nc82) (DSHB, University of Iowa, USA)
- 1:200 Rat α-FLAG Tag (NBP1-06712SS) (Novus Biologicals, Littleton, CO, USA)
- 1:300 Rabbit α-HA Tag (C29F4) (Cell Signalling Technologies, Danvers, MA, USA)

The following secondaries were used:

- 1:300 ATTO 647N Goat α-Rat IgG (612-156-120) (H and L) (Rockland, PA, USA)
- 1/1000 Cy3 Goat α-Rabbit (111-165-144) (Jackson Immuno Research, West Grove, PA, USA)
- 1/600 Cy2 Goat α-Mouse (115-225-146) (Jackson Immuno Research, West Grove, PA, USA)

Fixation dehydration and mounting was the same as protocol one above. Central brains were imaged using a Plan-Apochromat 63x/1.40 oil immersion objective (voxel size = 0.19 × 0.19×0.38 μm; 1024 × 1024 pixels). Tiles of regions of interest were stitched together into the final image (*Yu and Peng, 2011*).

## Protocol 3: MCFO

The following primaries were used:

- 1:30 Mouse α-Brp (nc82) (DSHB, University of Iowa, USA)
- 1:200 Rat α-FLAG Tag (NBP1-06712SS) (Novus Biologicals, Littleton, CO, USA)
- 1:300 Rabbit α-HA Tag (C29F4) (Cell Signalling Technologies, Danvers, MA, USA)
- 1:500 Mouse α-V5 (DL550) (AbD Serotec, Raleigh, NC, USA)

The following secondaries were used:

- 1/400 AF488 Goat α-Mouse (A-11001) (Life Technologies, now Thermofisher Waltham, MA, USA)
- 1:300 ATTO 647N Goat α-Rat IgG (612-156-120) (H and L) (Rockland, PA, USA)
- 1/500 AF594 Donkey α-Rabbit (A-21207) (Jackson Immuno Research, West Grove, PA, USA)

Fixation dehydration and mounting was the same as protocol one above, while imaging was the same as protocol two with specific parameters as described (*Nern et al., 2015*).

## Protocol 4: Neurotransmitter staining (VGlut and ChAT)

For protocol 4–5, the dissections, immunohistochemistry, and imaging of fly central nervous systems were done as previously described (*Aso et al., 2014a*). In brief, brains and VNCs were dissected in Schneider's insect medium and fixed in 2% paraformaldehyde (diluted in the same medium) at room temperature for 55 min. Tissues were washed in PBT (0.5% Triton X-100 in phosphate buffered saline) and blocked using 5% normal goat serum before incubation with antibodies.

The following primaries were used:

- 1/1600 Chicken α-GFP (ab13970, Abcam, UK)

- 1/100 Mouse α-ChAT (ChAT4B1, Hybridoma bank University of Iowa, USA)
- 1/500 Rabbit α-DVGlut (gift from Dion Dickman)

The following secondaries were used:

- 1/800 AF488 Goat α-Chicken (A-11039) (Life Technologies, now Thermofisher Waltham, MA, USA)
- 1/400 AF568 Goat α-Mouse (A-11031) (Life Technologies, now Thermofisher Waltham, MA, USA)
- 1/400 Cy5 Goat α-Rabbit (111-175-144) (Jackson Immuno Research, West Grove, PA, USA)

Fixation dehydration and mounting was the same as protocol one above. Imaging was performed with a Plan-Apochromat 63x/1.40 oil immersion objective (voxel size = 0.19 × 0.19×0.38 μm; 1024 × 1024 pixels) and neurotransmitter was determined by manually examining overlap with axonal boutons.

## Protocol 5: Neurotransmitter staining (GABA)

See protocol four for details on initial steps.
   The following primaries were used:

- 1/200 Rabbit α-GABA (A2052, Sigma-Aldrich, USA)
- 1/1600 Chicken α-GFP (ab13970, Abcam, UK)

The following secondaries were used:

- 1/800 AF488 Goat α-Chicken (A-11039) (Life Technologies, now Thermofisher Waltham, MA, USA)
- 1/400 AF568 Goat α-Rabbit (A-11011) (Life Technologies, now Thermofisher Waltham, MA, USA)

Fixation dehydration and mounting was the same as protocol one above, while imaging was performed with an EC Plan-Neofluar 40x/1.30 oil objective with 768 × 768 pixel resolution at each 1 μm, 0.6–0.7 zoom factor.

## Light microscopy image processing and analysis

All image analysis was performed using Fluorender (*Wan, 2012*), macros written with Fiji (http://fiji.sc/) and data was imported in R (https://www.r-project.org/) for data analysis and visualisation. Image registration was carried out as described (*Jefferis et al., 2007*; *Cachero et al., 2010*). Registration was based on confocal images where the neuropil is stained with the nc82 antibody. All brains were registered either to JFRC2 (*Jenett et al., 2012*), FCWB (*Chiang et al., 2011*) or JFRC2013 (*Aso et al., 2014a*). For the assignment of expression patterns to different brain regions, a standard brain with label fields was used to name different brain regions (*Ito et al., 2014*).

   For the creation of masks for light microscopy analysis (*Figures 2–8*) we first performed segmentations in Fluorender. These were mostly performed on polarity images but for four lines, the different genomic position of the polarity effectors led to broader patterns and instead a high-resolution Chrimson::mVenus stain or a MCFO image with full expression (rather than single neurons) was used (imaged in the same manner as polarity, see above). In all cases we removed background and off-target neurons by segmentation. We then separated neurons into their majority axon and dendrite, using the Syt::smHA puntate stain, although most neurons were not totally polarized.

   For each category of segmentation (whole neuron, dendrite-only, axon-only) we created a mask from their different samples by overlaying n = 3 examples of each line. This was followed by contrast enhancement, gaussian blurring (sigma = 1) and auto thresholding (Otsu method) to create a mask. All image analysis was performed using Fiji. For local neurons we could not assign axon and dendrite at light-level (if it existed), therefore we segmented all of the neuron except for primary neurite and soma (termed membrane).

   Overlap comparisons for pairs of masks (eg. LHON dendrite and LHIN axon) were compared in R (https://www.r-project.org/) using the cmtk.statistics (https://www.nitrc.org/projects/cmtk) function in the 'nat' package (*Costa et al., 2016*). For two masks (eg. mask1 and mask2), we first calculated the percentage overlap mask1 has with mask2 (P1). Next we calculated the percentage overlap mask2 has with mask1 (P2). The reported value was the mean of P1 and P2. We also calculated

overlap by taking the maximum and minimum of the P1 and P2 values with similar results (data not shown). Hierarchical agglomerative clustering (by euclidean distance and complete linkage) and heatmap generation was performed with the 'NMF' R package.

For analysis of projection neurons (PNs) we used a combined dataset of different PNs from several published sources (*Jefferis et al., 2007*; *Yu et al., 2010*; *Grosjean et al., 2011*; *Lin et al., 2013*; *Chiang et al., 2011*), registered to the JFRC2013 by bridging registrations (*Manton et al., 2014*).

For V glomerulus PNs we used a subset of well-registered PNs from the Flycircuit database (*Chiang et al., 2011*; *Lin et al., 2013*). As several V glomerulus PN cell types did not have established nomenclature we expanded the nomenclature of a previous study (*Lin et al., 2013*). See *Table 1* below.

Where possible we assigned different sensory modalities to LHINs (*Figure 7*). For taste and temperature input neurons we used published data (*Kim et al., 2017*; *Frank et al., 2015*) or clear arborization in the SEZ (*Wang et al., 2004*). For temperature data we named the temperature PNs according to their tract in published work. For WED-PN1-5, we assigned them as mechanosensory based on a recent paper examining the function of the wedge in mechanosensation and potentially wind-sensing (*Patella and Wilson, 2018*). For two LH cell types that had dendrites in the fly optic lobes, we assigned these as visual (*Zhu, 2013*). For creating averaged LHIN modality representations (*Figure 7D–E*) we took an exemplar of each LHIN with an assigned modality and averaged pixel values across sensory modalities using Fiji. Only pixels in the LH were displayed.

For analysis with MBONs and DANs we used previously published registered data (*Aso et al., 2014a*) to create axon and dendrite masks respectively.

MCFO image data was semi-automatically converted into single cell neuronal skeletons (as represented by the SWC file format (*Cannon et al., 1998*) using FEI Amira 6.4.0. Each registered stack was split into its constituent colour channels, a threshold for pixel inclusion wa manually set and where single cells were determined to be separable, neurons were skeletonised using the Filament tool. We regularly observed off-target single cell labelling, as effector site differed to the site used for screening (attP18). Careful annotation of MCFO data (in 'lhns' package) was critical prior to morphological clustering (see below).

Skeletonised neurons could be directly compared with other neuronal skeletons related to the lateral horn that have recently been collated by *Frechter et al., 2019*, including a large number of MCFO derived skeletons from the FlyCircuit database (*Chiang et al., 2011*). In order to determine the cell types present in our dataset, we used NBLAST (*Costa et al., 2016*) to compare our MCFO data with the library of named cell types built by *Frechter et al., 2019*. In a minority of cases, we could not find a match and a 'new' cell type was added to the Frechter et al. naming scheme.

## Sparse Electron Microscopy reconstruction and neuron identification

We leveraged a newly available ssTEM image volume for a single, whole adult female *Drosophila melanogaster* fly brain (*Zheng et al., 2018*), which had been imaged at x,y,z resolution 4 nm x 4 nm x 40 nm. Neuronal skeleton reconstruction was performed according to the strategies of (*Schneider-Mizell et al., 2016*), using CATMAID (http://www.catmaid.org), a Web-based environment for working on large image datasets. We aimed to identify lateral horn cell types for which we had generated

**Table 1.** Table of V Glomerulus PN designations.
Name refers to their use in this paper, Flycircuit ID refers to the neuron ID in the Flycircuit database (http://www.flycircuit.tw/) and their corresponding figure (*Lin et al., 2013*).

| Name | Flycircuit ID | Figure in *Lin et al., 2013* |
| --- | --- | --- |
| PN-V1 | VGlut-F-2000095 | Figure $1A_d$ |
| PN-V5 | VGlut-F-000463 | Figure $1C_a$ |
| PN-V6 | VGlut-F-800016 | Figure $S2B_a$ |
| PN-V7 | VGlut-F-200269 | Figure $2D_a$ |
| PN-V8 | VGlut-F-600026 | Figure $S2A_a$ |

DOI: https://doi.org/10.7554/eLife.43079.021

MCFO or other image data in this single high-resolution volume using two broad methods. (1) By identifying anatomical loci in the EM that corresponded with anatomical features for our identified lateral horn neurons, for example, their primary neurite tracts. We bridged between loci identified at light-level and the EM (*Manton et al., 2014*) in order to build a list of candidate neurons. (2) We used NBLAST to search for our MCFO derived morphologies against extant lateral horn tracing in this dataset. Over the past ~2 years a community of researchers across the world have been reconstructing neurons in the *Zheng et al., 2018* volume, allowing us to, with their consent, NBLAST against thousands of partial reconstructions to build upon our candidate list for each cell type (see acknowledgements). Candidates where then traced until they diverged from the morphologies we expected from our light-level data. For reasons of time, morphologies that remained consistent with our expectations were reconstructed until the end of all their microtubule-containing branches (*Schneider-Mizell et al., 2016*). Microtubules could easily be identified by eye in the electron micrographs. Neurons are not expected to produce large divergent arbours from neurite that does not contain microtubules. Reconstructing the microtubule-less 'twigs' of a neuron is time consuming for human tracers, and would be prohibitively time consuming given the number of neuronal cell type we sought to identify. NBLAST was used to confirm that our final targets were more similar to the cell type were were searching for, than other cell types in our light-level library. In a few exemplar cases, we completed neurons fully, including pre- and postsynapse annotation as described in other CATMAID-based *Drosophila* connectomic studies, for example (*Zheng et al., 2018*)(*Zheng et al., 2018*; *Dolan et al., 2018*). In total, we estimate that we and our acknowledged contributors spent ~700 hr reconstructing neurons and ~130 hr reviewing neurons to ensure they had been reconstructed correctly.

R tools for accessing the CATMAID API are available on github by following links from jefferislab. org/resources. The catmaid and elmr R packages provide a bridge between a CATMAID server and the R statistical environment and bridging registration tools respectively. They include several add-on packages from the NeuroAnatomy Toolbox (nat see http://jefferis.github.io/nat/) suite enabling statistical analysis and geometric transformation of neuronal morphology. Further analysis relied on unreleased custom R code developed by A.S.B and G.S.X.E.J.

## Optogenetic stimulation of LH neurons: Valence Assay

We used a four quadrant assay (*Takemura et al., 2017*; *Aso et al., 2014b*) while recording the behaviour of groups of flies (*Figure 9A*). Note we used two different assays, the newer version (*Aso and Rubin, 2016*) was used for all valence screen experiments with the exception of *Figure 9D*, which used the previous iteration (*Aso et al., 2014b*).

Behaviour was performed essentially as described (*Aso et al., 2014a*). For each vial of experimental flies, between 3–5 each of males and females were crossed on normal fly food supplemented with 1/500 all-trans-retinal (Sigma-Aldrich, MO, USA). Parental flies were allowed oviposit for 2–4 days before being tipped into a new vial, to control for population density of offspring. All lines were grown at 22 degrees in a covered box in a 12 hr light:dark cycle incubator.

Upon eclosion females of the correct genotype were sorted on a cold plate and kept on fresh 1/ 250 all-trans-retinal food. Experiments were generally performed less than 3 days after cold sorting but occasionally later. In this case the flies were tipped into a fresh vial of 1/250 all-trans-retinal food. All sorted females were again stored at 22 degrees in a covered box in a 12 hr light:dark cycle incubator. For behaviour only 3–7 day old flies were used. All experiments were performed in the dark at 25°C with 50–55% relative humidity.

For the valence assay, groups of 20 female flies were used for each individual experiment and n refers to the number of experiments. As previous studies had observed airflow dependent phenotypes for olfactory sensory neuron stimulation (*Bell and Wilson, 2016*), we maintained a flow rate of clean air from each of the four arms at 100 mL/min. This was controlled by mass-flow controllers, and air was extracted from the central hole at 400 mL/min. Note the previous old design (used in *Figure 9D*) did not have precisely controlled airflow but did have similar extraction.

The choice assay was performed in a circular arena previously described (*Aso and Rubin, 2016*). The dimensions were 10 cm diameter and 3 mm high. The arena was divided into four sections, and each quadrants could be illuminated with 617 nm LEDs (Red-Orange LUXEON Rebel LED—122 lm; Luxeon Star LEDs, Brantford, Ontario, Canada). Each trial is two minutes long. A single trial consists of two 30 s OFF periods where the area is not illuminated. Two 30 s ON periods are interspersed

between the OFF periods. For each on period a separate pair of quadrants are stimulated. Video recordings were performed under IR backlight using a camera (ROHS 1.3 MP B and W Flea3 USB 3.0 Camera; Point Grey, Richmond, BC, Canada) with an 800 nm long pass filter. Videos were analysed using custom Fiji scripts (http://fiji.sc/) and R (https://www.r-project.org/) with the 'tiff' package (see https://github.com/jefferislab/flywatch). The preferences index (PI) was calculated as: [(number of flies in Q2 and 3) - (number of flies in Q1 and 4)]/(total number of flies). The single value PI was calculated by averaging the PIs from the final 5 s of each of the two red-light ON segments of the paradigm.

To initially examine the changes in non-valence parameters, we examined the behaviour of flies in our valence assay during optogenetic stimulation, mining the same dataset collected for valence. For the calculation of delta metric (eg. Delta pixel per second, delta degrees per second and delta distance from center), the first five second window during the red-light ON stimulation was used for analysis and only animals that remained in the lit quadrants (*Figure 9A*) for the full window were used. To calculate the delta metric we took the difference in mean metric before and during stimulation and normalized this to the mean metric.

## Optogenetic stimulation of LH neurons: Whole-Field Stimulation in Flybowl

For full field stimulation we used the Flybowl (*Simon and Dickinson, 2010*; *Wu et al., 2016*), as this allowed for homogenous stimulation and ensured flies did not crowd at the area edges. The same fly handling protocols were used as for valence assay. For this assay, each fly was considered as a single experimental n, as flies were homogeneously illuminated (*Sen et al., 2017*). Unlike the valence assays, there is no airflow in this assay.

For the activation of neurons expressing ChrimsonR, the arena was uniformly illuminated with 617 nm LEDs (Red-Orange LUXEON Rebel LED - 122 lm; Luxeon Star LEDs, Brantford, Canada). The three stimulation intensities used were 3.59 (weak), 11.88 (medium) and 25.2 (strong) uW/mm$^2$ and each were five seconds long. Videos were recorded under reflected IR light using a camera (ROHS 1.3 MP B and W Flea3 USB 3.0 Camera; Point Grey, Richmond, Canada) with an 800 nm long pass filter at 30 frames per second. Videos were analysed using custom Fiji scripts (http://fiji.sc/) and R (https://www.r-project.org/) with the 'tiff' package.

## Optogenetic stimulation of LH neurons: Fixed, flying flies

A blue LED circular arena was assembled with 44 panels (4 rows and 11 columns, spanning 330 degrees in azimuth and 120 degrees in elevation) as described (*Reiser and Dickinson, 2008*), with the LED emission peaking at 464 nm (Bright LED Electronics Corp., BM-10B88MD). Two layers of blue filter (Roscolux #59, Stamford US) were laid on top of the LED panels to allow 0.04% transmission. Each fly was tethered at the end of a tungsten wire as described and positioned in the center of the arena (*Reiser and Dickinson, 2008*). An 880 nm LED (Digi-Key, PDI-E803-ND) illuminated the fly from above. A custom-built wingbeat analyzer (University of Chicago Electronics Shop, US) measured the wingbeat frequency and amplitudes for both wings. Yaw turning was computed as the left minus right wingbeat amplitude. Thrust was computed as the sum of the left and right wingbeat amplitudes. In a dark trial, all LED panels were off. In a stationary blue stripe trial, a blue stripe of 4 pixels (15 degrees) wide and 32 pixels (120 degrees) tall was presented directly in front of the fly throughout the duration of the trial. In a closed-loop blue stripe trial, the same blue stripe counter-rotated relative to the direction of the fictive yaw turning, with a gain of −10, so that the fly had control of the yaw position and rotational velocity of the stripe. A pair of fiber-coupled 660 nm activation LEDs (Thorlabs, M660F1) were positioned above the fly and aimed at the head. The light intensity at the head was 200 mW/mm$^2$.

A computer (Dell R5500, US) controlled the timing of the experiments through a data acquisition card (National Instruments, USB-6229 BNC, US) and sampled the flight parameters at 1 kHz. Each fly went through six blocks (repetitions) of trials, with each trial randomly interleaved within the block. Each trial was a combination of visual (dark, stationary blue stripe, closed loop blue stripe) and optogenetic LED stimulation (no stim, stim) conditions, and lasted 10 s. Inter-trial interval was 20 s. Stimulation started after 5 s in each trial, with the activation LEDs driven at 2.5 V and 40 Hz for 500 ms (20 pulses, 12 ms pulse width).

The effects of optogenetic activation were computed as the change of flight parameters induced by the LED stimulation. The average measured in the 100 ms before stimulation was first subtracted from all measurements, the mean stimulated time trace for each fly and each visual condition was then computed by subtracting the mean time trace of each flight parameter over the six trials without LED stimulation from that with LED stimulation.

## Statistics

All statistics was carried out in R. For optogenetic valence assays, groups of 20 female flies were used for each 'n'. For Flybowl and fixed, flying animal experiments n represents the number of animals due to homogenous stimulation. For all experiments, previous published work was used to determine the appropriate sample size (*Reiser and Dickinson, 2008*); *Aso et al., 2014b*; *Sen et al., 2017*).

For the Split-GAL4 optogenetic valence screen (*Figure 9—figure supplement 1A*), a Levene test indicated that our data was homoskedastic. We performed a Kruskal-Wallis rank sum test followed by a Dunn's post-hoc test (dunn.test.control from the 'PMCMR' R package) comparing the control performance index (Empty Split-GAL4) against that of each Split-GAL4 line. P-values were adjusted to control the False Discovery Rate (*Abramovich and Benjamini, 2006*) at 10% and declared significant (blue in *Figure 9—figure supplement 1A*) for False Discovery Rate-adjusted $p<0.05$. *Figure 9B* displays only the significantly different Split-GAL4 lines. This same procedure was used to determine lines that had statistically different locomotion, turning or movement upwind from control within the same dataset.

For replication experiments derived from the larger optogenetic valence screen (ie. *Figure 9D and G*) and for forward locomotion Flybowl experiment (*Figure 10D*), we again performed a Kruskal-Wallis rank sum test followed by a Dunn's post-hoc test (dunn.test.control from the 'PMCMR' R package) comparing the control performance index (Empty Split-GAL4) against that of each Split-GAL4 line. P-values were adjusted by Bonferroni correction.

For experiments in fixed, flying animals, at each time point after LED on, the mean stimulated data across n = 6 flies with CsChrimson expression was compared with those without CsChrimson using Wilcoxon rank sum test, statistical significance was labelled on each plot.

## Data presentation

Most graphs were generated using the open source R package ggplot2 and related packages. Plots of fixed, flying animal behaviour were generated in MATLAB (Mathworks, USA).

## Data availability

All split-GAL4 lines in *Supplementary file 1* (i.e. the best lines for each cell type) are available to order from Janelia (http://splitgal4.janelia.org/cgi-bin/splitgal4.cgi). Lines used in non-screen behavioural experiments (e.g. *Figure 9D,G* and *Figure 10*) are also available on this website and the additional lines for each cell type (*Figure 1* and *Supplementary file 2*) will be distributed upon request.

Image data (screening image and where available MCFO and/or polarity) for the best two lines for each cell type (where available, *Supplementary file 1*) can be downloaded (www.janelia.org/split-gal4). Segmented single neurons (MCFO) presented in *Figures 2–5*, light microscopy segmentations of examples of each cell type analysed and EM data is available in the 'lhns' R package (https://github.com/jefferislab/lhns), which is viewable at http://jefferislab.org/si/lhlibrary. See key resource table for listing of all github links. Raw optogenetic behaviour videos can be made available upon request.

## Code availability

All R packages described above are available by following links from http://jefferislab.org/resources and github (https://github.com/jefferis/nat). Packages include full documentation and sample code. Custom scripts used to generate figures are available on github (*Dolan et al., 2019*; copy archived at https://github.com/elifesciences-publications/2018lhsplitcode).

## Acknowledgements

This work was supported by MRC LMB Graduate Studentships, Boehringer Ingelheim Fonds PhD Fellowships (M-JD and ASB), the Herchel Smith scholarship (ASB) and a Janelia Graduate Research Fellowship (M-JD); ERC Starting (211089) and Consolidator (649111) grants and core support from the MRC (MC-U105188491) (to GSXEJ); the Howard Hughes Medical Institute (YA, DB and GMR); and a Wellcome Trust Collaborative Award (203261/Z/16/Z to GSXEJ, DB, GMR). This work was also supported by the Janelia Visiting Scientist Program. The FlyLight Project Team (led by Oz Malkesman) performed brain dissections, histological preparations and confocal imaging for the Split-GAL4 characterisation, Polarity and MCFO data. The Janelia Project Technical Resources team (KC, CC, led by GI) performed brain dissections and histological preparations for neurotransmitter data. We thank Dion Dickman for his kind gift of the VGlut antibody. We thank the Janelia Drosophila Resources team, especially Todd Laverty, Karen Hibbard, Jui-Chun Kao, Amanda Cavallaro and Brandi Sharp for assistance with crosses, sorting and expert advice. We thank Shiuan-Tze Wu for assistance with behavioural screening. We thank Vivek Jayaraman for comments on the manuscript and sharing the 20xUAS-ChrimsonR stock. We thank Claire Managan, Amelia Edmondson-Stait, Kimberly Meechan, Nadiya Sharifi, Markus Pleijzier, Jeremy Johnson, Arian Jamasb, Philipp Ranft, Najla Masoodpanah, Corey Fisher, Steven Calle, Joseph Hsu, Jacob Ratliff, Istvan Taisz, Zane Mitrevica, Addy Adesina, and Arlo Sheridan for collectively contributing a total of 28% of the EM reconstruction work. Finally we thank the Jefferis and Rubin groups (especially Tanya Wolff), Gwyneth Card and Michael Winding for many insightful comments on the manuscript.

## Additional information

### Funding

| Funder | Grant reference number | Author |
|---|---|---|
| Howard Hughes Medical Institute | Janelia Graduate Research Fellowship | Michael-John Dolan |
| Boehringer Ingelheim Fonds | PhD Fellowship | Michael-John Dolan<br>Alexander Shakeel Bates |
| Medical Research Council | LMB Graduate Studentships | Michael-John Dolan<br>Alexander Shakeel Bates |
| Herchel Smith Scholarship | | Alexander Shakeel Bates |
| Wellcome | 203261/Z/16/Z | Davi D Bock<br>Gerald M Rubin<br>Gregory SXE Jefferis |
| Howard Hughes Medical Institute | | Davi D Bock<br>Yoshinori Aso<br>Gerald M Rubin<br>Gregory SXE Jefferis |
| Medical Research Council | MC-U105188491 | Gregory SXE Jefferis |
| European Research Council | Starting grant 211089 | Gregory SXE Jefferis |
| European Research Council | Consolidator grant 649111 | Gregory SXE Jefferis |

The funders had no role in study design, data collection and interpretation, or the decision to submit the work for publication.

### Author contributions

Michael-John Dolan, Conceptualization, Resources, Data curation, Software, Formal analysis, Supervision, Funding acquisition, Validation, Investigation, Visualization, Methodology, Writing—original draft, Project administration, Writing—review and editing; Shahar Frechter, Conceptualization, Investigation, Data curation, Validation, Project Administration; Alexander Shakeel Bates, Conceptualization, Data curation, Formal analysis, Investigation, Visualization, Writing—original draft, Writing—review and editing; Chuntao Dan, Investigation, Formal analysis, Visualization; Paavo Huoviala, Ruairí JV Roberts, Serene Dhawan, Christina Christoforou, Kari Close, Ben Sutcliffe, Bianca Giuliani,

Investigation; Philipp Schlegel, Investigation, Visualization, Supervision; Remy Tabano, Investigation, Visualization; Heather Dionne, Resources, Investigation; Feng Li, Supervision, Investigation, Visualization; Marta Costa, Geoffrey Wilson Meissner, Supervision, Investigation; Gudrun Ihrke, Supervision, Validation, Resources; Davi D Bock, Resources, Supervision, Funding Acquisition; Yoshinori Aso, Conceptualization, Supervision, Investigation, Methodology, Software, Writing—review and editing, Funding Acquisition; Gerald M Rubin, Conceptualization, Supervision, Funding acquisition, Writing—original draft, Writing—review and editing; Gregory SXE Jefferis, Conceptualization, Software, Supervision, Funding acquisition, Investigation, Writing—original draft, Project administration, Writing—review and editing

## Author ORCIDs

Michael-John Dolan 
Shahar Frechter 
Alexander Shakeel Bates 
Chuntao Dan 
Philipp Schlegel 
Ben Sutcliffe 
Feng Li 
Marta Costa 
Geoffrey Wilson Meissner 
Davi D Bock 
Yoshinori Aso 
Gerald M Rubin 
Gregory SXE Jefferis 

## Decision letter and Author response

Decision letter https://doi.org/10.7554/eLife.43079.028
Author response https://doi.org/10.7554/eLife.43079.029

## Additional files

### Supplementary files

• Supplementary file 1. All LH cell types presented in this study. A table of each cell type listed in this study. Where we could not separate cell types genetically we grouped them together as one (see text). For each cell type we display the best two lines (where available), the polarity (Output, Input or Local), the neurotransmitter (acetylcholine, GABA and/or glutamate), the average number of neurons in these lines, if that LH cell type is a 'core' LH cell type (see discussion) and, where available, a match to the cell types in *Jeanne et al., 2018*.
DOI: https://doi.org/10.7554/eLife.43079.022

• Supplementary file 2. All split-GAL4 lines presented in this study. A table of all split-GAL4 lines listed by their line code, hemidrivers, constituent cell types and line quality (good, ideal or combinatorial, see text). Note that combinatorial lines with multiple cell types have an entry for each cell type.
DOI: https://doi.org/10.7554/eLife.43079.023

• Supplementary file 3. All split-GAL4 lines screened in optogenetic valence assay. A table of split-GAL4 lines are listed by their line code, presence of a significant valence response in our assay (see Materials and methods for statistics) and cell type. Note the screen included three control genotypes: empty split-GAL4, Gr66a (aversion) and MB83c (attraction).
DOI: https://doi.org/10.7554/eLife.43079.024

• Supplementary file 4. Summary MCFO data for PV4a1:5 single cell labelling. Different lines for PV4a1:5 which had a behavioural phenotype were used to generate single neurons by the MCFO approach (*Nern et al., 2015*). Each split-GAL4 line and the number of neurons of each cell type (eg. PV4a2) are listed.
DOI: https://doi.org/10.7554/eLife.43079.025

• Transparent reporting form
DOI: https://doi.org/10.7554/eLife.43079.026

## Data availability

All split-GAL4 lines in Supplementary file 1 (i.e. the best lines for each cell type) are available to order from Janelia (http://splitgal4.janelia.org/cgi-bin/splitgal4.cgi). Lines used in non-screen behavioural experiments (e.g. Figure 9D,G and Figure 10) are also available on this website and the additional lines for each cell type (Figure 1 and Supplementary file 2) will be distributed upon request. Image data (screening image and where available MCFO and/or polarity) for the best two lines for each cell type (where available, Supplementary file 1) can be downloaded (www.janelia.org/split-gal4). Segmented single neurons (MCFO) presented in Figures 2–5, light microscopy segmentations of examples of each cell type analysed and EM data is available in the 'lhns' R package (https://github.com/jefferislab/lhns), which is viewable at http://jefferislab.org/si/lhlibrary. See key resource table for listing of all github links. Raw optogenetic behaviour videos can be made available upon request.

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
