## [Decision Letter]

Thank you for submitting your article "Neurogenetic dissection of the *Drosophila* innate olfactory processing center" for consideration by *eLife*. Your article has been reviewed by two peer reviewers, and the evaluation has been overseen by a Reviewing Editor and K VijayRaghavan as the Senior Editor. The following individual involved in the review of your submission has agreed to reveal her identity: Ilona C Grunwald Kadow (Reviewer #2).

The reviewers have discussed the reviews with one another and the Reviewing Editor has drafted this decision to help you prepare a revised submission.

Summary:

With their new set of split GAL4 lines labeling specific types of lateral horn neurons in the *Drosophila* brain, Dolan et al. have created a hugely important resource for the field of *Drosophila* olfaction, which parallels the similarly important pair of *eLife* papers from Aso et al. in 2014 describing mushroom body output neurons. Not only have Dolan et al. created and screened thousands of split GAL4 lines, but they have shown the usefulness of the novel genetic tools resulting from this impressive feat, by analyzing the anatomy, connectivity, and behavioral function of the lateral horn neurons that they now have genetic access to thanks to their new split GAL4 lines. The manuscript is well-written, clear and thorough, and in particular, does a good job of organizing a potentially unwieldy set of results into a few major findings:

1) LH output neurons project mainly to the SLP

2) Non-olfactory inputs innervate the ventral, not dorsal, LH

3) Links between LH and MBONs

4) Behavioral effect of activating some LH neurons

This work has been a huge effort and we would like to congratulate the authors on this great achievement. The availability of these lines will certainly help to finally unravel what the LH can and cannot do. Therefore, the work should be published as soon as possible to allow access and boost research on the LH.

Essential revisions:

1) In what sense can neurons be classified as LHINs and LHONs if they are incompletely polarized? Is this a qualitative (subjective?) judgment based on the amount of Syt::HA signal in the LH vs. elsewhere? Is the incomplete polarization what explains why some LHONs have axons in the LH? (Figure 4—figure supplement 1B) This should be clarified.

2) Paragraph five subsection “Interactions between Lateral Horn Output Neurons and Mushroom Body neurons”, AD1b2 connectivity with MBON. Were there zero synapses from AD1b2 onto MBONs? It's not clear from the text if they looked for AD1b2->MBON synapses and found none, or if they just didn't look for any. This should be clarified.

3) The clustering in Figure 5—figure supplement 1A’ suggests there is something special about uniglomerular PNs from DA3, vVL1, and the newly named V1, V5, V6, V7, V8. DA3 and VL1 seem to be severely under-represented in the inputs to KCs according to Caron et al., 2013 (and most PNs from the V glomerulus skip the mushroom body altogether). Could the authors comment or speculate on what this might mean?

4) PD2a1/b1 neurons are required for innate attraction for some odors (Dolan et al., 2018), but here, optogenetically activating them with LH991 in the quadrant assay causes aversion. Is this experimental noise / a false positive? Could activity in the same neurons cause attraction in one context and avoidance in another? Is this group of 6-7 cells still functionally heterogeneous? e.g. some go toward approach and some go toward avoidance? Similarly, what are we to make of the finding that there aren't anatomical labeled lines between "attractive" and "aversive" PNs and specific LH neurons? (e.g. AV1a1 gets both attractive and aversive inputs, but activating AV1a1 causes avoidance.) The authors already have a nuanced discussion in subsection “Functions of the Lateral Horn” of behavioral outputs of LH neurons and discussing these issues would fit there nicely.

5) In addition to the impressive anatomical work, Dolan et al. also present a number of results using different behavioral assays. The results are interesting, but difficult to make sense of in respect to their recent results and results by the lab of Rachel Wilson regarding odor categorization. I find the negative results almost more informative than the positive hits because they suggest that the LH might somehow drive more specific behaviors than analyzed and that there is yet a lot to discover, which is another reason to publish this work swiftly and allow the community to contribute.

6) The author's argument that the LH drives innate behavior rests in part on the observation that activation of LH output neuron results in expression of attraction, aversion or more specific motor behavior. The same regarding valence has been demonstrated for output neurons of the MB as noted by the authors (see for instance Aso et al., 2014). Inactivation of KC or MBON affects odor responses of naïve flies (see for instance Owald et al., 2015; Lewis et al., 2015). MBONs show some segregated responses toward innately positive vs. negative odors (Hige et al., 2015). What exactly are the arguments for the classification of the LH as a center for innate behavior, in particular as supposed to the MB? One argument is the hardwiring as compared to the MB, but then again see Iurilli and Datta, Neuron 2017. In addition, if animals can learn valence using the MB, how do they learn category without the LH? I believe that it might be sensible to formulate conclusions more carefully to leave more room for future experimental designs and findings. It is likely that MB and LH do different things, but I am not convinced that 'innate' vs. 'learning' is the best description. By contrast, the authors beautifully show the interconnections between MB and LH as well as the specificities of the LH, which could be used to revolutionize the view on the LH rather than to emphasize old dogmas.

---

## [Author Response]

Essential revisions:1) In what sense can neurons be classified as LHINs and LHONs if they are incompletely polarized? Is this a qualitative (subjective?) judgment based on the amount of Syt::HA signal in the LH vs. elsewhere? Is the incomplete polarization what explains why some LHONs have axons in the LH? (Figure 4—figure supplement 1B) This should be clarified.

We thank the reviewers for pointing this out. We have more clearly defined this below and in the Materials and methods section.

Our categorization of LHON and LHIN is based two criteria:

1) A qualitative examination of the Syt::HA stain in the LH versus other regions. Although we did see incomplete polarisation (see Figure 1—figure supplement 1 for example), it was typically weighted heavily in one direction.

2) A qualitative examination of synapses in our traced single neurons reconstructed in the EM dataset; a quantitative examination would require completely reconstructed neurons, which is prohibitively time consuming for the scale of this study.

3) In one case, there was a discrepancy between these two approaches and we favoured the conclusion from EM data. Specifically, ell-type PV2d1 appears to have the majority of its presynaptic signal in the LH (Figure 4—figure supplement 1B). However, examination of presynapses in EM strongly implied these are actually LHONs. This discrepancy may be due to overexpression of Syt::HA or ectopic synapses due to over expression of Syt itself.

We have added the above details to the Materials and methods section “Split-GAL4 line characterization (Polarity, Single neurons, Neurotransmitter)” to more clearly explain our categorization.

2) Paragraph five subsection “Interactions between Lateral Horn Output Neurons and Mushroom Body neurons”, AD1b2 connectivity with MBON. Were there zero synapses from AD1b2 onto MBONs? It's not clear from the text if they looked for AD1b2->MBON synapses and found none, or if they just didn't look for any. This should be clarified.

We thank the reviewers for pointing out this omission. There are very few connections in this direction, only 2 MBONs get synapses from AD1b2: MBON-y2a’1 with 2 synapses and MBON-y5B’2a with 6 synapses. None for MBON-a1/a’2. We have added this detail to the Results section; subsection “Interactions between Lateral Horn Output Neurons and Mushroom Body neurons”: paragraph six.

3) The clustering in Figure 5—figure supplement 1A’ suggests there is something special about uniglomerular PNs from DA3, vVL1, and the newly named V1, V5, V6, V7, V8. DA3 and VL1 seem to be severely under-represented in the inputs to KCs according to Caron et al., 2013 (and most PNs from the V glomerulus skip the mushroom body altogether). Could the authors comment or speculate on what this might mean?

We agree that it is very interesting that many CO2 PNs do not project to the calyx. This likely reflects the special ethological value CO2 has for behaving flies. For DA3 and VL1, it is more difficult to speculate as less is known about the odors that stimulate these cells. We might speculate that a similarly ecologically-relevant odor ligand may exist that stimulates DA3 and VL1. As this is speculatory and several studies have already discussed CO2 encoding, we feel discussion of this is interesting but beyond the scope of our paper.

4) PD2a1/b1 neurons are required for innate attraction for some odors (Dolan et al., 2018), but here, optogenetically activating them with LH991 in the quadrant assay causes aversion. Is this experimental noise / a false positive? Could activity in the same neurons cause attraction in one context and avoidance in another? Is this group of 6-7 cells still functionally heterogeneous? e.g. some go toward approach and some go toward avoidance? Similarly, what are we to make of the finding that there aren't anatomical labeled lines between "attractive" and "aversive" PNs and specific LH neurons? (e.g. AV1a1 gets both attractive and aversive inputs, but activating AV1a1 causes avoidance.) The authors already have a nuanced discussion in subsection “Functions of the Lateral Horn” of behavioral outputs of LH neurons and discussing these issues would fit there nicely.

We thank the reviewer for pointing out this discrepancy. In the case of the PD2a1/b1, as the reviewer points out, only one line (L991) had a phenotype in the optogenetic arena. We believe this is due to off-target expression in the VNC as three other PD2a1/b1 split-GAL4 lines did not exhibit a gain of function phenotype and a different line was attractive in our flight assay. The reviewer is right in that we cannot exclude different subsets of a cell type being labelled however. We have addressed this by adding the above point to the discussion of advantages and limitations of our work in subsection “Advantages and limitations” paragraph three.

For the second point, we have now confirmed a subset of these PN-to-LHON connections for the DA2 glomerulus with EM reconstruction in a preprinted paper from Huoviala et al. which address these points for this case: https://www.biorxiv.org/content/biorxiv/early/2018/08/26/394403.full.pdf. More generally however, these predictions will need to be validated to determine if this is a common rule. This is especially the case for light microscopy overlap data, which underwent multiple registrations (see below).

5) In addition to the impressive anatomical work, Dolan et al. also present a number of results using different behavioral assays. The results are interesting, but difficult to make sense of in respect to their recent results and results by the lab of Rachel Wilson regarding odor categorization. I find the negative results almost more informative than the positive hits because they suggest that the LH might somehow drive more specific behaviors than analyzed and that there is yet a lot to discover, which is another reason to publish this work swiftly and allow the community to contribute.

We thank the reviewers for pointing out the usefulness of this behavioral data. We agree that many different assays and contexts will be needed to pick apart the differing functions of the LH (see also Discussion). We also agree that one take home message of this paper is that the LH and its interactions are more complicated than previously thought. Our behavioral work did not reveal a clear unifying principle for LHONs (in contrast to the classification of MBONs into spatially separate groups of opposite valence, eg Aso et al., 2014); we suspect that this is a real reflection of diverse of behavioral roles for LHONs. Of course optogenetic activation experiments have their caveats (also mentioned in the Discussion). Higher resolution assays (moving away from valence), loss-of-function experiments with a diversity of odors and behavioral states will, in our opinion, be key to cracking the LH circuitry. We have tried to convey these points clearly in the Discussion.

The reviewers also ask about recent work from the Wilson lab (Jeanne et al. 2018), which provides clear functional evidence that LHNs show biased pooling of input PNs. Jeanne et al. propose that particular combinations of input PNs might define “salient olfactory scenes”. However, we could not yet identify any obvious relationships between the behavioral results for different cell types and the reported inputs. Comprehensive EM reconstruction of all the neurons that pass information from AL to LH will certainly help. To facilitate future investigation of this point, we included a relationship between the neurons in our lines/EM reconstructions and those recorded in Jeanne et al. in Supplementary file 1.

6) The author's argument that the LH drives innate behavior rests in part on the observation that activation of LH output neuron results in expression of attraction, aversion or more specific motor behavior. The same regarding valence has been demonstrated for output neurons of the MB as noted by the authors (see for instance Aso et al., 2014). Inactivation of KC or MBON affects odor responses of naïve flies (see for instance Owald et al., 2015; Lewis et al., 2015). MBONs show some segregated responses toward innately positive vs. negative odors (Hige et al., 2015). What exactly are the arguments for the classification of the LH as a center for innate behavior, in particular as supposed to the MB? One argument is the hardwiring as compared to the MB, but then again see Iurilli and Datta (Neuron 2017). In addition, if animals can learn valence using the MB, how do they learn category without the LH? I believe that it might be sensible to formulate conclusions more carefully to leave more room for future experimental designs and findings. It is likely that MB and LH do different things, but I am not convinced that 'innate' vs. 'learning' is the best description. By contrast, the authors beautifully show the interconnections between MB and LH as well as the specificities of the LH, which could be used to revolutionize the view on the LH rather than to emphasize old dogmas.

We completely agree that our gain-of-function experiments *alone* do not demonstrate a role for the LH in innate behavior. Furthermore, we certainly take the reviewer’s point that dividing the LH and MB into innate versus learned centres is (of course) an oversimplification. Nevertheless, we do think that this remains a useful generalisation given recent data about the LH and MB (including from this paper, the companion Frechter et al. manuscript available as a bioRxiv preprint, and other recent published and preprinted work from our group and others). In particular, many exceptions to this general rule can be explained by lateral interactions between LHONs, DANs and MBONs:

1) As the reviewer points out: PNs wire stereotypically with LH neurons and this is reflected in their olfactory responses (Jefferis et al., 2007, Fisek et al., 2015, Jeanne et al., 2018, Frechter et al., 2018). This is in contrast to PN-to-MB connections which are pesudorandom or random (Gruntman and Turner 2013, Caron et al., 2013).

2) Neuronal silencing experiments demonstrate that abolition of KC neurotransmission leads to a wiping of memory and a reset to innate olfactory responses (Heimbeck et al., 2001, Parnas et al., 2013).

3) Studies rescuing neuronal plasticity genes in null backgrounds consistently point to the MB by providing a full phenotypic rescue (Mao et al., 2004, McGuire et al., 2003, Zars, 2000).

4) Recent work from our lab has demonstrated that specific LH neurons are required for innate olfactory attraction in the T maze (Dolan et al., 2018) and for egg-laying preference (Huoviala et al., 2018, bioRxiv), two innate olfactory behaviours. Although PD2a1/b1 LHNs mediate memory recall, this seems best understood as the modulation by a mushroom body output pathway of an “innate”, hardwired pathway that plays a role in naive behaviour (Dolan et al., 2018).

5) In this paper, we show many interactions between LH and MB. For categorical learning, we would propose that LHON/DAN interactions may mediate this behaviour. Similarly, although individual MBONs may have preferential physiological responses to odors of attractive and negative valence, this could likely also be explained by LHON-to-DAN interactions in the presynaptic MB compartment.

6) As the reviewer points out, the MB has also shown to be implicated in innate behaviours (Bräcker et al., 2013, Lewis et al., 2015). This has however, been limited to modulation by internal state for CO2 processing, which has an elaborate set of diverse, atypical PNs (Lin et al., 2013).

7) With respect to Iurilli and Datta, Neuron 2017, they identified many neurons in the olfactory amygdala that respond to both aversive and appetitive stimuli, concluding that their responses are not organised according to valence. In our behavioural screen, very few LH neurons seem to drive a clear valence response. Our LM overlap data suggests that LHONs can integrate PNs that are innately aversive and attractive (see also Huoviala, 2018 preprint for an EM resolution study focusing on an aversive pathway). In this sense, the Iurilli results and our own are quite consistent. To us, those results do not challenge a role in innate olfactory behaviour, but rather suggest that we need to move beyond attraction/aversion when examining innate olfaction.

In contrast Iurilli et al. find no evidence for stereotyped responses across animals in the olfactory amygdala whereas electrophysiological recordings in Frechter et al. show that this is the case for the *Drosophila* lateral horn. While this may well represent a design difference between insects and mammals, we do note that they recorded neurons at random without regard to e.g. genetically defined cell type. Had we performed similar random sampling in the LH, blind to the massive diversity of cell types, we would probably have missed the response stereotypy that we now report in Frechter et al. We include an extended discussion of this paper in the Frechter et al. manuscript.

Overall, we definitely agree that the time is ripe to form a new and more nuanced synthesis of the relative roles of different olfactory brain areas. The bulk of these data (nearly all now cited in the text, see subsection “Advantages and limitations”) suggest – to the extent that individual brain areas can be considered in isolation – that the LH is involved is oriented to innate behaviour but that there is much cross-talk between LH and MB.